# VISUAL PERCEPTUAL TO CONCEPTUAL FIRST-ORDER RULE LEARNING NETWORKS

## ABSTRACT

Learning rules plays a crucial role in deep learning, particularly in explainable artificial intelligence and enhancing the reasoning capabilities of large language models. While existing rule learning methods are primarily designed for symbolic data, learning rules from image data without supporting image labels and automatically inventing predicates remains a challenge. In this paper, we tackle these inductive rule learning problems from images with a framework called $\gamma$ILP, which provides a fully differentiable pipeline from image constant substitution to rule structure induction. Extensive experiments demonstrate that $\gamma$ILP achieves strong performance not only on classical symbolic relational datasets but also on relational image data and pure image datasets, such as Kandinsky patterns.

## 1 INTRODUCTION

Automatically learning rules is becoming increasingly important with the development of artificial intelligence. The learned rules serve as interpretable representations that enable systems to generalize better (Liu et al., 2023; Xie et al., 2025) , and provide transparent explanations for the input data (Kaur et al., 2023; Gao et al., 2025). Beyond propositional rules, first-order rules allow one to describe properties of and relations between constants at a general level; such expressiveness is highly demanded in trustworthy applications (Dwivedi et al., 2023). In the first-order rule learning domain, most existing methods (Gao et al., 2024; Hocquette et al., 2024; Cropper & Muggleton, 2016) are designed for learning from relational symbolic data. Despite their efficiency, the growing availability of multimodal data makes learning rules from knowledge graphs with image constants (Cunnington et al., 2023; Shindo et al., 2023) increasingly important.

However, a challenge for inductive rule learning from *relational image* domains is *symbol grounding* without label leakage: The inability to ground visual inputs to symbolic variables in formal systems without explicit supervision (Topan et al., 2021; Harnad, 1990). Hence, when inductively constructing rules from image inputs, existing methods are considered to have access to the label information of image constants (Evans et al., 2021; Evans & Grefenstette, 2018; Shindo et al., 2023), which is regarded as *label leakage*. In this paper, we assume that image symbolic labels are neither required nor leaked during inductive learning, reducing human effort and enabling fully automated rule learning from raw data. Moreover, the absence of relational descriptions for target events often necessitates introducing new predicates, a fundamental challenge known as *predicate invention* in *inductive logic programming* (ILP) (Muggleton & Buntine, 1988; Kok & Domingos, 2007).

In this paper, we propose a novel inductive rule learning framework, $\gamma$ILP, which learns rules from image-based constants with both predefined constant relations (e.g., relational image data) and implicit or undefined constant relations (e.g., Kandinsky image data). When learning from data without relations, we further create suitable concepts as relations in the learned rules for describing image instance classes. The proposed method is fully differentiable: It takes constant embeddings for neural networks as input and learns rules through analyzing the parameters of the well-trained neural networks. In more detail, we use a pre-trained encoder to embed the image constants and relations when they are defined. In case relations are missing, we first generate rules using predicate placeholders. We interpret the semantics of the predicate placeholder by analyzing the image constants represented by the variables and the order of the variables from the output of $\gamma$ILP. Furthermore, we employ multimodal LLMs to translate the semantics of these placeholder predicates into natural language format, thereby capturing the relations between constants.

Briefly summarized, the main contributions of this work are: Firstly, we develop an inductive reasoning process that is fully differentiable and operates in latent space, where constant substitution and rule structure induction are performed via tensor operations on GPUs. Secondly, we present the $\gamma$ILP framework for learning rules from relational image data without symbolic image constant labels, avoiding label leakage and enabling symbolic grounding. Thirdly, we tackle predicate invention by analyzing the learned image constants represented by variables in the learned rules, and utilize LLMs as translators to generate symbolic predicate semantics.

To the best of our knowledge, $\gamma$ILP is the first providing all features from above. Our experiments show strong performance of $\gamma$ILP not only on classical symbolic relational datasets, but also on relational image data and pure image dataset Kandinsky patterns (Müller & Holzinger, 2021).

**Organization.** We review related work on rule learning in Sec. 2, followed by preliminaries on logic programs, ILP, and encoder architectures in Sec. 3. In Sec. 4, we present the proposed method, including the knowledge base generator, differentiable substitution mechanism, and predicate invention tasks. We present experimental results in Sec.5, conclusions and future works in Sec.6, and the code in: drive.google.com/drive/folders/10x-TXo2nJuoZTPKDz-sbybgBnC-Rvcwo?usp=sharing.

## 2    RELATED WORK

**ILP methods.**    Inductive logic programming (ILP) was introduced by Muggleton (1991) to induce rules that combined with background knowledge derive positive examples. Symbolic ILP methods (Cropper & Dumancic, 2022) typically adopt top-down strategies (e.g., FOIL (Quinlan, 1990)), bottom-up approaches (e.g., CIGOL (Muggleton & Buntine, 1988)), or hybrids like Aleph (Srinivasan, 2001) to discover logical rules. These systems are not integrated with neural networks for scalable learning using GPUs. Learning from interpretation transition (Inoue et al., 2014) is an ILP framework that learns propositional rules from input-output pairs, which has been integrated into neural networks (Gao et al., 2022b). Baugh et al. (2023; 2025) proposed a neural network to learn propositional rules to describe multiclass data. The challenge here lies in learning first-order rules. To leverage GPU computation, Evans & Grefenstette (2018) proposed $\partial$ILP, which learns rules from symbolic inputs using logic templates in differentiable operations. DFORL (Gao et al., 2024) learns first-order rules from symbolic data via bottom-up propositionalization (França et al., 2014), but its non-differentiable process on the substitution prevents end-to-end training with the rule learning network. NeurRL (Gao et al., 2025) extends this network to learn rules from raw time series in a differentiable way, yet it overlooks relations between raw image data (Evans & Grefenstette, 2018). $\gamma$ILP learns rules in a bottom-up way without any pre-defined logic templates in a fully differentiable way from ground substitution to rule induction.

**Symbol Grounding.**    $\alpha$ILP (Shindo et al., 2023) induces logic programs from visual inputs, comprising a trained perception module and a symbolic fact converter. Cunnington et al. (2024) replaced the converter with an LLM, while (Evans & Grefenstette, 2018; Evans et al., 2021) used predicted symbolic image labels for differentiable reasoning models. Wang et al. (2019) applied neural networks to solve maximum satisfiability with image symbolic labels. All these approaches rely on image symbolic labels as reasoning module inputs. In satisfiability, Topan et al. (2021) emphasized symbol grounding, and showed that we cannot achieve the expected performance without explicit supervision. Aware of this, $\gamma$ILP induces rules from images without symbolic labels but their representations, preventing label leakage.

**LLMs and ILP.**    Creswell & Shanahan (2022); Han et al. (2024) discussed the deduction reasoning abilities using LLMs under natural languages. Li et al. (2025) test the inductive reasoning abilities of LLMs on observed facts, which are not formally described in first-order language. Under the ILP setting, de Souza et al. (2025) propose a systematic methodology to analyse the ILP capabilities and limitations of LLMs. We further test the ILP abilities of the LLMs with the state-of-the-art reasoning abilities. Gentili et al. (2025) utilizes LLMs to rename the predicate placeholder with natural language semantics solely based on the provided logic rules with predicate placeholders. However, $\gamma$ILP invents the semantics of relations by analyzing the learned constants represented by variables, and we utilize LLMs to translate these semantics to a natural language format.

## 3 PRELIMINARIES

### 3.1 LOGIC PROGRAMS

We consider a *first-order* language $L = (R, F, C, V)$ (Lloyd, 1984), where $R$, $F$, $C$, and $V$ denote (countable) sets of predicate symbols, function symbols, constants, and variables, respectively. A *term* $t$ is a constant, a variable, or an expression $f(t_1, \ldots, t_n)$, where $f$ is an $n$-ary function symbol and $t_1, \ldots, t_n$ are terms. An *atom* is of the form $p(t_1, \ldots, t_n)$, where $p$ is an $n$-ary predicate symbol. A *literal* is an atom or its negation. A *clause* is a finite disjunction of literals. A *rule* (or *definite clause*) is a clause with exactly one positive literal and can be written as: $\alpha_0 \vee \neg\alpha_1 \vee \cdots \vee \neg\alpha_n$ or equivalently in implication form as: $\alpha_0 \leftarrow \alpha_1, \alpha_2, \ldots, \alpha_n$, where $\alpha_0$ is called the *head* of the rule (denoted head$(r)$), and $\{\alpha_1, \ldots, \alpha_n\}$ is the *body* (denoted body$(r)$). Each $\alpha_i$ in the body is referred to as a *body atom*. Variables in the head atom are *head variables*; those only in the body are *auxiliary variables*. A *fact* is a rule with an empty body. A *logic program* $P$ is a set of rules.

In first-order logic, a term, atom, clause, etc. is *ground* if it contains no variables. A *substitution* is a finite set $\theta = \{V_1/t_1, \ldots, V_n/t_n\}$, where each $V_i$ is a distinct variable and each $t_i$ is a term different from $V_i$. A *ground substitution* includes only ground terms. For an atom $\alpha$, the expression $\alpha\theta$ denotes the ground atom obtained by applying a ground substitution $\theta$ to the variables in $\alpha$. Additionally, the set of all ground instances of rules in a logic program $P$ is denoted as ground$(P)$. The *Herbrand base* $B_P$ of a logic program $P$ is the set of all ground atoms constructable from the predicate symbols and constants in $P$. An *interpretation* is a subset $I \subseteq B_P$ that contains the ground atoms regarded as true. The semantics of $P$ is based on the *immediate consequence operator* (van Emden & Kowalski, 1976; Lloyd, 1984) $T_P : 2^{B_P} \to 2^{B_P}$ which is defined as $T_P(I) = \{\,\text{head}(r) \mid r \in \text{ground}(P), \text{ body}(r) \subseteq I\,\}$.

### 3.2 INDUCTIVE LOGIC PROGRAMMING

In the setting of learning from entailments (Muggleton & De Raedt, 1994; Evans & Grefenstette, 2018), a specific ILP learning task seeks to generate a logic program $P$ that derives a goal concept represented by a *target predicate* $p_t$, given a tuple $(B, \mathcal{P}, \mathcal{N})$. Here, $B$ denotes a set of ground atoms called *background knowledge*, and $\mathcal{P}$ and $\mathcal{N}$ are sets of ground atoms $p_t(c_1, \ldots, c_n)$ representing true instances (*positive examples*) and false instances (*negative examples*), respectively. A logic program $P$ is a *solution* of $(B, \mathcal{P}, \mathcal{N})$ if $B \cup P$ entails all positive examples in $\mathcal{P}$ and none of the negative examples in $\mathcal{N}$. An atom with the target predicate $p_t$ is called a *target atom*.

In *propositional* logic, each atom amounts to a Boolean variable. When learning propositional logic programs (Inoue et al., 2014), an interpretation $I$ contains Boolean values of all atoms that appear in the body of any rule in the logic program, while another interpretation $J$ contains the Boolean values of the head atoms. The learned logic program $P$ satisfies $T_P(I) = J$ for all pairs $(I, J) \in E$, where $E$ is a set of interpretation pairs.

Gao et al. (2022a) used propositionalization to build interpretation transitions through grounding all possible body atoms and the target atom with substitutions for learning first-order logic programs. An input interpretation vector $\mathbf{x}$ represents the Boolean values of all possible non-ground body atoms under a substitution $\theta$, and the output $y$ represents the Boolean values of the target atoms under $\theta$. Based on this, Gao et al. (2025) proposed NeurRL, a neural architecture that learns the immediate consequence operator $T_P$ and extracts a first-order logic program $P$ from its trained parameters, ensuring that $P$ satisfies the ILP learning setting. The neural network is designed as follows:

$$\hat{y} = \text{RuleNetwork}(\mathbf{x}) = \widetilde{\bigvee}(f_m \cdot f_{m-1} \cdots f_1(\mathbf{x})), \tag{1}$$

where $\hat{y}$ is the predicted Boolean value of the target atom, the right-hand side uses the fuzzy disjunction operator $\widetilde{\bigvee}(x_1, \ldots, x_n) = 1 - (1 - x_1) \cdot \ldots \cdot (1 - x_n)$, and the $i$-th layer $f_i$ is:

$$f_i(\mathbf{x}) = \frac{1}{1 - d}\text{ReLu}(\mathbf{M}_i\mathbf{x} - d),$$

with $d$ as the fixed bias. To interpret neural networks into a set of rules, the sum of weights connected to the same hidden node in the next layer is constrained to 1 in each layer (Gao et al., 2024). Hence,

the softmax activation function is applied on each row of every trainable layer $\tilde{\mathbf{M}}_i$:

$$\mathbf{M}_i[j,k] = \frac{e^{\tilde{\mathbf{M}}_i[j,k]}}{\sum_{u=1}^{n_{\text{in}}} e^{\tilde{\mathbf{M}}_i[j,u]}}, j \in [1, n_{\text{out}}], k \in [1, n_{\text{in}}],$$

where $n_{\text{in}}$ and $n_{\text{out}}$ denote the number of columns and rows of $\tilde{\mathbf{M}}_i$, respectively. After training, rules are extracted from the *logic program tensor* $\mathbf{M}_P = \prod_{i=1}^m \mathbf{M}_i$; each row corresponds to a rule, and elements in the row correspond to atoms, with atoms exceeding a threshold included in that rule.

*Generalization* (Plotkin, 1970; Buntine, 1988) applies a substitution to replace specific terms, such as constants, with more general ones, such as variables, within a logic program. Through it, the rules can be regarded as knowledge that is applicable to a wider range of examples.

Throughout the paper, let $\mathbb{E}$ represent the image constant embeddings under the background knowledge $B$, and let $\mathbb{Z}$ represent a latent space. The symbol grounding problem targets establishing a mapping from an input $\mathbf{e} \in \mathbb{E}$ to some latent state $\mathbf{z} \in \mathbb{Z}$ that is fed into a predefined symbolic reasoning procedure for producing the final output $y$. The training data contains only the image constants $\mathbf{e}$'s and the corresponding $y$'s (Li et al., 2023). The labels of latent states $\mathbf{z}$ are not leaked, putting the problem into a weakly-supervised setting (Zhou, 2017). Predicate invention (Kok & Domingos, 2007) refers to the discovery of new concepts, properties, and relations from data, expressed in terms of observable predicates.

### 3.3 ENCODERS

Encoders transfer the raw data into embeddings. Vision transformers (ViT) (Dosovitskiy et al., 2021) generate the embeddings for images. A variational autoencoder (VAE) (Kingma & Welling, 2014) is a type of generative model in deep learning that learns to encode data into a compressed latent representation and then decode it back to reconstruct the original data.

Clustering methods group similar embeddings into the same clusters. In this work, we adopt the differentiable clustering approach proposed by Fard et al. (2020). Let $K$ be the number of clusters, $\mathbf{c}_i \in \mathbb{R}^d$ represent the $i$-th cluster center, where $d$ is the embedding dimension, and $\mathcal{C} = \{\mathbf{c}_1, \mathbf{c}_2, \ldots, \mathbf{c}_K\}$ denote the set of cluster representations. Then the clustering objective is defined as:

$$L_{\text{cluster}} = \sum_{\mathbf{e} \in \mathbb{E}} \sum_{i=1}^K f(h(\mathbf{e}), \mathbf{c}_i) \cdot G_{i,f}(h(\mathbf{e}), \alpha; \mathcal{C}), \tag{2}$$

where $h$ is the encoder function, $f$ is a distance metric (e.g., mean squared error), and G is a differentiable weighting function assigning maximum weight to the minimal distance (Jang et al., 2017):

$$G_{i,f}(h(\mathbf{e}), \alpha; \mathcal{C}) = \frac{\exp(-\alpha f(h(\mathbf{e}), \mathbf{c}_i))}{\sum_{i'=1}^K \exp(-\alpha f(h(\mathbf{e}), \mathbf{c}_{i'}))},$$

with $\alpha > 0$. Larger $\alpha$ makes $G$ closer to the discrete minimum, while smaller $\alpha$ smooths training.

## 4 METHOD

We propose $\gamma$ILP, a fully differentiable ILP framework depicted in Figure 1, that learns first-order logic programs from relational image data or Kandinsky patterns, where explicit relations are undefined. Learning involves generalizing from image constants to cluster indices, then constructing non-ground atoms to describe target atoms or image classes. $\gamma$ILP consists of a deep clustering module serving as a generalization function, a latent knowledge base generator, and a rule learning neural network with a novel differentiable substitution method. The output of $\gamma$ILP is a logic program which, in case the relations between the constants are not well-defined, will contain predicate placeholders, and the semantics of predicate placeholders can be inferred from images represented by the variables. Additionally, $\gamma$ILP incorporates with LLMs to obtain the semantics of the predicate placeholders in symbolic format.

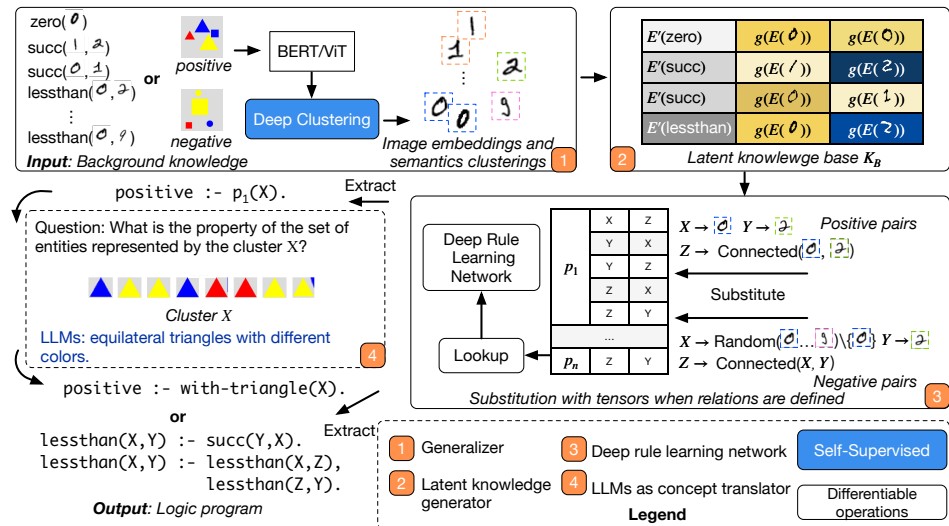

Figure 1: The pipeline of the learning framework. $E$ and $E'$ indicate the encoder function for image and text, respectively.

## 4.1 DEEP CLUSTERING MODULE AND KNOWLEDGE BASE GENERATOR

In the paper, each constant is in image format, and relations are predicates. When relations $r$ between image constants $e$ are predefined, each instance in a background knowledge $B$ is represented as $r(e_1, e_2)$, and the data is called relational image data. We induce the logic programs to describe the target atom. When such relations are undefined but are essential for characterizing the target atom, we set each image instance as background knowledge $B$, and each object inside the image instance is regarded as an image constant. The data is regarded as pure image data, and the representative benchmark is Kandinsky patterns. We induce the logic program $P$ based on $B$ to describe the image class, and the variables in $P$ can be substituted with the image object constants.

In ILP, the generalization function $g$ replaces specific terms with generalized terms, e.g., constants with variables. Clustering groups similar constants under a centroid, and the clustering serves as a generalization function $g : \mathbb{E} \rightarrow \mathcal{C}; \mathbf{e} \mapsto \mathbf{c}_i$, where $\mathbf{c}_i$ is the centroid containing the image constant $\mathbf{e}$. For adaptively learning the clusters of constants, we use the differentiable one defined in Eq. (2). Then, we transfer $B$ to a *latent knowledge base* denoted as $K_B$:

$$K_B = \begin{cases} \{\, \mathbf{r} \oplus g(\mathbf{e}_1) \oplus g(\mathbf{e}_2) \mid r(e_1, e_2) \in B\}, \text{when relations are defined} \\ \{g(\mathbf{e}) \mid e \in B\}, \text{ when relations are not defined} \end{cases},$$

where $\oplus$ indicates the concatenation between vectors, $\mathbf{e}$ and $\mathbf{r}$ denote the embeddings of constant $e$ and relation $r$ by encoders, respectively.

## 4.2 DIFFERENTIABLE SUBSTITUTION

The differentiable substitution method produces substitutions at the batch level. In the sequel, we confine to unary and binary predicates, but the method can be extended to predicates of arbitrary arity. Let $N$ be the dimension of the input $\mathbf{x}$ and $d$ the number of variables in the learned logic program. We describe the substitution methods for both the defined and undefined relations as follows.

**Relations are predefined.** Let $R_b$ and $R_u$ denote the sets of binary and unary relations, respectively. When the relations are defined, $N = |R_b| \times P(d, 2) + |R_u| \times d - 1$, where $P(d, 2) = d(d-1)$ is the number of permutations of $d$ distinct elements taken two at a time. The subtraction of 1 indicates that the target atom is excluded from being considered as a possible body atom. Algorithm 1 outlines the differentiable substitution procedure when relations are defined, where positive and negative substitution sets ($\Theta^+$, $\Theta^-$) are constructed for supervised learning with labeled data. If

---

**Algorithm 1** Differentiable substitution method

---

**Input**: Variables $X(V_1), Y(V_2), V_3, \ldots, V_d$ and $d \geq 1$; the binary or unary target atom $p_t(X, Y)$ resp. $p_t(X)$; the background knowledge $B$; and the set $\mathbb{E}$ of all constant embeddings.
**Output**: Positive substitution set $\Theta^+$ and negative substitution set $\Theta^-$.

1: Initialize the substitution sets $\Theta^+$ and $\Theta^-$ as empty.
2: Update the clustering module and get centroid embedding $g(\mathbf{e})$ for each constant embedding $\mathbf{e} \in \mathbb{E}$.
3: **while** batch size is not reached **do**
4:     Initialize two substitutions $\theta^+$ and $\theta^-$ as empty sets.
5:     **if** $p_t$ is a binary predicate **then**
6:         Randomly select the constant pair $(\mathbf{e}_x, \mathbf{e}_y)$ for a positive example $p_t(e_x, e_y) \in B$, and another embedding $\mathbf{e}_x^- \in \mathbb{E} \setminus \{\mathbf{e}_x\}$.
7:         Add $X/g(\mathbf{e}_x), Y/g(\mathbf{e}_y)$ to $\theta^+$ and $X/g(\mathbf{e}_x^-), Y/g(\mathbf{e}_y)$ to $\theta^-$.
8:     **else**
9:         Randomly select the constant embedding $\mathbf{e}_x$ for a positive example $p_t(e_x) \in B$, and another embedding $\mathbf{e}_x^- \in \mathbb{E} \setminus \{\mathbf{e}_x\}$.
10:        Add $X/g(\mathbf{e}_x), Y/g(\text{Random}(\mathbb{E}))$ to $\theta^+$ and $X/g(\mathbf{e}_x^-), Y/g(\text{Random}(\mathbb{E}))$ to $\theta^-$.
11:    **end if**
12:    Randomly choose the constant representations $\mathbf{e}_3^+, \ldots, \mathbf{e}_d^+, \mathbf{e}_3^-, \ldots, \mathbf{e}_d^- \in \text{Random}(\mathbb{E})$.
13:    Add $V_3/g(\mathbf{e}_3^+), \ldots, V_d/g(\mathbf{e}_d^+)$ to $\theta^+$ and add $V_3/g(\mathbf{e}_3^-), \ldots, V_d/g(\mathbf{e}_d^-)$ to $\theta^-$.
14:    Add the substitution $\theta^+$ and $\theta^-$ to $\Theta^+$ and $\Theta^-$, respectively.
15: **end while**
16: **return** $\Theta^+$ and $\Theta^-$.

---

an instance $r(e_1, e_2)$ exists in $B$, we consider the embeddings $\mathbf{e}_1$ and $\mathbf{e}_2$ to be *connected*. We define the function $\text{Connected}(\mathbf{X}, \mathbf{Y})$ to retrieve all constant embeddings connected to the constant embeddings $\mathbf{X}$ and $\mathbf{Y}$. The random selection function, denoted as $\text{Random}(\mathbb{E})$, returns a randomly chosen element from all image embeddings $\mathbb{E}$. For each substitution in $\Theta^+$, we substitute each head variable with the constant embeddings corresponding to the constant pair that appears in the positive examples. The auxiliary variables are then replaced with randomly selected embeddings from $\mathbb{E}$. For each substitution in $\Theta^-$, we assign the head variables to embeddings of the constant pair not present in the positive examples by replacing the variable $X$ with a random embedding. The auxiliary variables are similarly replaced with randomly selected embeddings from $\mathbb{E}$. In addition, when the target predicate $p_t$ is binary and the number of variables $d = 3$, the auxiliary variable $V_3$ in the rules connects the head variables $X$ and $Y$ following a forward-chaining pattern (Kaminski et al., 2018). This introduces a language bias of the form: $p_t(X, Y) \leftarrow p_1(X, V_3), p_2(V_3, Y)$. Note that the variables in the body atoms of this forward-chaining bias can be interchanged. Consequently, we replace the random function in Line 12 of Algorithm 1 with $\mathbf{e}_1 \in \text{Connected}(\mathbf{e}_x, \mathbf{e}_y)$, selecting embeddings that satisfy the forward-chain pattern to enhance the learning process.

**Relations are undefined.** When relations are not explicitly defined, the possible body atoms consist of one assigned predicate placeholder for each term list. Then, $N = C(d, 2) + d - 1$, where $C(d, 2) = d(d-1)/2$ is the number of combinations of $d$ elements taken two at a time. In addition, each image instance corresponds to a knowledge base $B$, and the object represented in an image instance consists of the constant embedding set $\mathbb{E}$. We design the substitution set as $\Theta = \{\{V_1/g(\mathbf{e}_1), \ldots, V_d/g(\mathbf{e}_d)\} \mid \mathbf{e}_1, \ldots, \mathbf{e}_d \in \text{Random}(\mathbb{E})\}$. The substitution here refers to replacing variables with cluster centroids, which is regarded as the symbolic assignments for random image constants derived from the clustering-based generalization function $g$.

## 4.3 DIFFERENTIABLE RULE LEARNING PROCESS

Each substitution in the substitution set can be regarded as a tensor with constant embeddings corresponding to all variables in the learned $P$. Besides, each substitution generates a training example $(\mathbf{x}, y)$, where $\mathbf{x}$ encodes the Boolean values of all possible body atoms. When relations are defined, $y$ indicates the Boolean value of the target atom. Conversely, when relations are not defined, $y$ indicates the label of the image instance class. We present how we generate the training examples for the differentiable rule learning module based on each substitution as follows.

**Relations are predefined.** For each non-ground atom $\alpha$ with a predicate $r$ and term list $\alpha_T$, we concatenate the constant embeddings to build the embedding of the ground atom $\alpha\theta$ as follows:

$$\alpha\theta = \mathbf{r} \oplus V_{i_1}\theta \oplus V_{i_2}\theta \oplus \cdots \oplus V_{i_n}\theta, \ V_{i_j} \in \alpha_T, \theta \in \Theta^+ \cup \Theta^-. \tag{3}$$

Then, we determine the ground truth of $\alpha\theta$ using a *lookup function* $L$, where $L(\alpha\theta) = 1$ if $\alpha\theta \in K_B$, and $L(\alpha\theta) = 0$ otherwise. Given all possible body atoms $\alpha_1, \ldots, \alpha_N$ and based on each positive substitution $\theta^+ \in \Theta^+$, we generate the positive input $\mathbf{x}^+ = [L(\alpha_1 \theta^+), \ldots, L(\alpha_N \theta^+)]$ and its label $y^+ = 1$. Similarly, we also generate the negative input $\mathbf{x}^- = [L(\alpha_1 \theta^-), \ldots, L(\alpha_N \theta^-)]$ and its label $y^- = 0$ under each negative substitution tensor $\theta^- \in \Theta^-$. With the tensor operations, we can look up the ground truth values for all possible body atoms under a batch of substitutions in $\Theta^+$ and $\Theta^-$, generate training examples $(\mathbf{x}, y)$, and train rule networks on GPUs in parallel.

**Relations are undefined.** When relevant predicates are not present in the training data, $\gamma$ILP can still learn first-order rules with predicate placeholders. We introduce a *variable constraint*: the number of variables in the learned logic program is set equal to the number of clusters. As a result, *each variable denoted as $\tilde{V}$ corresponds to a specific group of similar constants*. Logic programs with variables under constraints are called *constrained logic programs*. Hence, we can use a function RetrieveConstants($\tilde{V}_i$) to retrieve the constants represented by $\tilde{V}_i$ from the constrained logic program. This enables us to interpret the predicate placeholders $p_i$ in atoms $p_i(\tilde{V}_1, ... \tilde{V}_n)$ by analyzing the constants under each constrained variable. Let $\alpha$ be a possible body atom and $\alpha_T$ be its term list. The lookup function $L(\alpha\theta)$ to obtain the Boolean value of the ground atom $\alpha\theta$ is defined as:

$$L(\alpha\theta) = L(\tilde{V}_{i_1}\theta) \wedge L(\tilde{V}_{i_2}\theta) \wedge \cdots \wedge L(\tilde{V}_{i_n}\theta), \ \tilde{V}_{i_j} \in \alpha_T, \theta \in \Theta,$$

which indicates that if all symbolic assignments of image constants substituting for all variables in $\alpha$ under $\theta$ are in $K_B$ simultaneously, then the Boolean value of the ground atom $\alpha\theta$ with the placeholder predicate is true; otherwise, it is false. We apply the lookup function to all body atom boolean values under a substitution, then we can obtain one training instance $\mathbf{x}$, and the label $y$ indicates the class of the image instance. For Kandinsky patterns, an image instance includes multiple image constants. In each epoch, the substitution grounds the variables into random constant representations in an image instance.

Overall, for the defined relations or undefined relations learning settings, the loss function $H$ can be summarized as follows:

$$H = \mathrm{MSE}(y, \mathrm{RuleNetwork}(\mathbf{x})) + \lambda \cdot L_{\mathrm{cluster}}, \tag{4}$$

where RuleNetwork($\mathbf{x}$) is defined in Eq. (1) and $L_{\mathrm{cluster}}$ in Eq. (2). We jointly train the rule learning network and clustering module, enabling simultaneous adjustment of generalized embeddings for constant embeddings and rule structures. The rules are extracted from the well-trained rule networks.

After training the model, the rules are extracted according to the logic program tensor $\mathbf{M}_P$. When the relations are not well-defined, we can induce the semantics of predicate placeholder in an atom $\alpha$ based on the constrained variable order and the constant under these clusters corresponding to the constrained variables.

As shown by (Gubelmann, 2024), LLMs can infer linguistic meaning based on their pre-trained knowledge without extra labels. We further utilize LLMs as a function QueryLLM(prompt, $C$) to translate from the semantics of predicate placeholders presented in constant images $C$ under their constrained variables to natural language semantics by a well-design prompt descried in Algorithm 2. Moreover, Algorithm 2 constructs the final logic program P (Line 11) by merging LLM-induced predicates into a generalized predicate with variables representing arbitrary constants.

## 5 EXPERIMENTAL RESULTS

Rules are evaluated by *precision* and *recall*: precision is the fraction of substitutions satisfying both the body and the head among those satisfying the body, and recall is the fraction of ground-truth positives correctly induced (Gao et al., 2024). Precision reflects the correctness of a rule or logic program in avoiding false positives, while recall reflects completeness in classifying all target labels by avoiding false negatives. We use the AdamW optimizer (Loshchilov & Hutter, 2019) to train

---

**Algorithm 2** Inducing semantics of predicate placeholders

---

**Input**: Constrained logic programs $P_t$ with predicate placeholders.
**Output**: The learned logic program $P$.

1: **for** each rule $r$ in $P_t$ **do**
2:    **for** each atom $\_(V_i, V_j)$ **do**
3:       $C_i, C_j$ = RetrieveConstants $(V_i, V_j)$.
4:       $\_ \leftarrow$ QueryLLMs("What is the relation between the two ordered sets of images?", $C_i, C_j$).
5:    **end for**
6:    **for** each atom $\_(V_i)$ **do**
7:       $C_i$ = RetrieveConstants $(V_i)$.
8:       $\_ \leftarrow$ QueryLLMs("What is the common property of the set of images?", $C_i$) .
9:    **end for**
10: **end for**
11: Generalize rules in $P_t$ using the induced predicate semantics to obtain the final logic program $P$.
12: **return** $P$.

---

$\gamma$ILP. We run $\gamma$ILP on classical ILP datasets (Evans & Grefenstette, 2018) with explicit constant and relation labels to assess its ILP capability, and compare $\gamma$ILP with $\partial$ILP and DFORL. At the same time, we validate the inductive learning abilities of LLMs (GPT-5 and Gemini 2.5 Pro) on the classical ILP datasets and compare their results with $\gamma$ILP. All non-LLM experiments were run on a Linux server (7 cores Intel 8362, 245 GB RAM, NVIDIA A100). The results show that Gemini 2.5 Pro learns correct (precision 1) and complete (recall 1) logic program, $\gamma$ILP learns correct rules with three variables efficiently, and GPT-5 correctly learns rules for a reduced number of input instances. Detailed results and average running times of $\gamma$ILP are given in Table 4 of Appendix A.

### 5.1 Reasoning on Relational Image Datasets

We evaluate $\gamma$ILP 's inductive reasoning ability on relational images using the benchmarks of Evans & Grefenstette (2018), with MNIST digits as constants and avoiding leaking their labels. We make the datasets by replacing the constants in the classical ILP datasets with two MNIST images of the corresponding label. One relational fact is used for training and another for testing. Relations describing image constants are in text format. We use pre-trained VAE as the encoder for relations. For the image constants, we use pre-trained ViT and VAE as the encoders for image constants, and the models are denoted as $\gamma$ILP-ViT and $\gamma$ILP-VAE, respectively. Since $\gamma$ILP is the first model to learn rules from relational image datasets without symbolic label leakage, and LLMs are also considered as the inductive rule learner without image label as inputs, we here compare $\gamma$ILP with state-of-the-art multimodal LLMs, including Gemini 2.5 Pro, GPT-o3, and GPT-5. To prevent label leakage when testing LLMs as symbolic solvers, we replace each relation with the same random string. We set $\lambda = 1$ and use 10 clusters (matching the digit classes). We retain only the rules with a precision of 1 in the learned logic program, and the average recall of the learned logic program over ten runs is reported in Table 1. To validate the inductive reasoning ability of LLMs without leaking relation semantics, we replace each relation with a random string. Results show that GPT-5 learns complete rules, while Gemini 2.5 Pro and GPT-o3 learn incorrect rules when relation semantics are hidden under some tasks. $\gamma$ILP learns complete rules except in the Fizz and Buzz datasets. Under the Fizz and Buzz datasets, the correct rules are: $fizz(X) \leftarrow fizz(Y), succ(Y, Z_1), succ(Z_1, Z_2), succ(Z_2, X)$ and $buzz(X) \leftarrow buzz(Y), succ(Y, Z_1), succ(Z_1, Z_2), succ(Z_2, Z_3), succ(Z_3, Z_4), succ(Z_4, X)$, respectively. Hence, the correct rules require at least 4 and 6 variables, respectively, which exceed the length $\gamma$ILP can effectively induce under the time limit.

We evaluate rule-learning ability of $\gamma$ILP on the temporal MNIST sequence task (Evans et al., 2021), equating annotated relations with unlabeled MNIST images as constants. In MNIST sequences, each image is assigned a positional index. Let $\texttt{succ}(e_1, e_2)$ denote that the label of image $e_2$ is the successor of the label of the image $e_1$, $\texttt{start}(e) = True$ indicate $e$ is the image at the first index, and $\texttt{before}_n(e_1, e_2)$ indicate the image $e_1$ precedes the image $e_2$ by $n$ indices. The input image sequence is 0, 5, 1, 5, 2, 5, 3, 5, 4, 5, 5, 5, ..., with training limited to the first 12 images (Figure 4, Appendix B). Since the Apperception Engine (Evans et al., 2021) requires a pre-trained MNIST model and a logic template, its rule format differs; thus, we present only the rules generated

| Model | Predecessor | Odd | Even | Lessthan | Fizz | Buzz |
|-------|------------|-----|------|----------|------|------|
| Gemini 2.5 Pro | **1.00** | **1.00** | 0.00 | 0.20 | 0.00 | 0.00 |
| GPT-o3 | **1.00** | 0.00 | **1.00** | **1.00** | 0.00 | **1.00** |
| GPT-5 | **1.00** | **1.00** | **1.00** | **1.00** | **1.00** | **1.00** |
| $\gamma$ILP-ViT | **1.00** | **1.00** | **1.00** | **1.00** | - | - |
| $\gamma$ILP-VAE | **1.00** | **1.00** | **1.00** | **1.00** | - | - |

Table 1: Recall of the learned logic program on relational image datasets. Best results are shown in bold. A '–' indicates that the rules were learned correctly but incompletely.

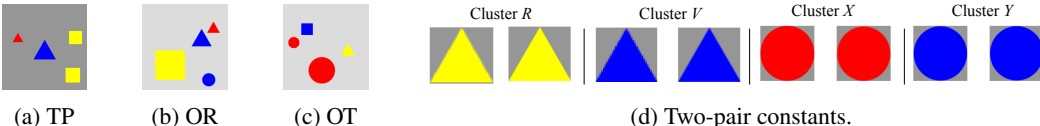

(a) TP      (b) OR      (c) OT                   (d) Two-pair constants.

Figure 2: Kandinsky patterns for tasks (a) two-pair (TP), (b) one-red (OR), and (c) one-triangle (OT). Constants represented by variables in the learned rules for TP are shown in (d).

by $\gamma$ILP. Let the index of the first image be 1. Assume an image $e$ has an even index and we set `target`$(e)$ to True. Then, a learned rule with the precision 1 when using VAE or ViT as encoders is the same as follows: `target`$(X) \leftarrow$ `before`$_8(X,Y) \wedge$ `before`$_{10}(X,Y) \wedge$ `target`$(Y)$. This captures either variable $X$ or $Y$ representing two images labeled 5 at different indices, and the distance between the two images is 2. Also, it captures regularities at distances 8 and 10 among images labeled 5. Then, assume an image $e$ has an odd index and we set `target`$(e)$ to True. The learned rule with precision and recall 1 by using ViT or VAE in $\gamma$ILP is the same as follows: `target`$(X) \leftarrow$ `succ`$(X,Y) \wedge$ `before`$_2(X,Y) \wedge$ `target`$(Y)$, stating that if two images are two indices apart, the latter's label succeeds the former's.

## 5.2 REASONING WITH PREDICATE INVENTION

We apply $\gamma$ILP to classify binary Kandinsky patterns (Müller & Holzinger, 2021) and assess its predicate invention ability without leaking constant labels. Each Kandinsky image instance includes multiple image objects as constants. Constant relations, though undefined in the instance, are essential for describing positive instances in first-order logic. Each constant in Kandinsky instances has a color (red, blue, yellow) and a shape (circle, square, triangle). Three patterns are illustrated in Figs. 2a–2c: two-pair (two disjoint object pairs of the same shape, one pair sharing color and the other differing), one-red (at least one red object), and one-triangle (at least one triangle-shaped object). We extract all non-grey subareas as image constants and learn first-order rules using placeholder predicates to represent the Kandinsky image instance class.

We used 30 instances per Kandinsky pattern for training and 30 for testing, with balanced positive and negative instances. Classification accuracy across models is shown in Table 2, including the CNN-based model ResNet (He et al., 2016), ViT, YOLO v5 (Redmon et al., 2016) with an MLP layer, and prominent LLMs. As we evaluate with LLMs, all learning strategies follow the few-shot strategy. Each experiment was run ten times, reporting the highest accuracy for best interpretability. Learned rules from baselines and the sensitivity of $\gamma$ILP are discussed in Appendices E and F, respectively.

When interpreting the learned rules, we found that both the ViT-based encoder and the VAE-based encoder can recover the correct rules under the best accuracy. For the two-pair task, we obtained two rules with predicate placeholders $p_1$ and $p_2$: `Positive` $\leftarrow p_1(V,R)$[1] and `Positive` $\leftarrow p_2(X,Y)$. Figure 2d shows the constants represented by the clusters $V$, $R$, $X$, and $Y$. The relation of constants under clusters $V$ and $R$ (or $X$ and $Y$) are the same shape but different colors. More generated rules are given in Appendix C. To translate the semantics in natural language of

---

[1]For simplicity, we rewrote the first-order rule `Positive`$(I) \leftarrow p_1(V,R) \wedge$ `Include`$(V,R,I)$, where the variable $I$ can be substituted by an image instance, including the image object substituting the variable $V$ and $R$.

| Task | CNN | ViT | YL | GM | G-o4 | G-4.1 | G-o3 | G-5 | $\gamma$ILP-VAE | $\gamma$ILP-ViT |
|------|-----|-----|-----|------|------|-------|------|------|-----------|-----------|
| Types | VM | VM | VM | LLM | LLM | LLM | LLM | LLM | RM | RM |
| TP | 0.50 | 0.63 | 0.40 | 0.46 | 0.56 | 0.47 | 0.50 | 0.67 | 0.64 | **0.75** |
| OR | 0.50 | 0.80 | 0.80 | **1.00** | 0.31 | 0.52 | 0.63 | 0.38 | 0.77 | **1.00** |
| OT | 0.50 | 0.90 | 0.80 | **1.00** | 0.53 | 0.40 | 0.37 | 0.78 | 0.77 | **1.00** |

Table 2: Classification accuracy on Kandinsky tasks: Two-pair (TP), one-red (OR), and one-triangle (OT). VM, RM, GM, YL, G, and G-o4 refer to vision models, reasoning models, Gemini 2.5 Pro, YOLO, GPT, and GPT-o4-mini.

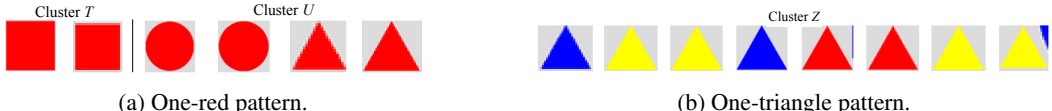

(a) One-red pattern.     (b) One-triangle pattern.

Figure 3: Constants represented by clusters for the one-red and one-triangle patterns.

placeholder predicates, we first randomly choose 20 constants from all constants. Then, we input constants under the constraints variables in the learn rules, along with a well-defined prompt. Specifically, the LLMs generated the semantics for predicates $p_1$ and $p_2$ as: "same shape (triangle) with different colors" and "same shape (circle) with different colors", respectively. Generalizing all predicate semantics induced by LLMs instructed by Line 11 in Algorithm 2, we obtain the final rule: `Positive` $\leftarrow$ `same_shape_and_different_color`$(X, Y)$, where $X$ and $Y$ denote any two constants in the image instance. The rule states that if two constants in an instance share the same shape but differ in color, the instance is a two-pair pattern. The rule achieves a recall of 1 but a precision below 1, as it ignores another pair of constants with the same color and shape.

For one-red, two learned rules are `Positive` $\leftarrow p_1(T)$ and `Positive` $\leftarrow p_2(U)$, with constants represented by $T$ and $U$ shown in Figure 3a. LLM-translated semantics for the placeholders are $p_1 =$ "shape in square, color in red" and $p_2 =$ "shape in circle or triangle, color in red". Generalizing these yields the final rule: `Positive` $\leftarrow$ `color_in_red`$(U)$. It means if any red constant occurs in an instance, then the instance is a one-red pattern. The precision and recall of the rule are both 1.

For the one-triangle task, a learned rule is `Positive` $\leftarrow p(Z)$, where the constants represented by $Z$ are shown in Figure 3b. LLM-translated semantics for the unary predicate $p$ is "all constants are in the shape of a triangle". Furthermore, we generalize the rule is `Positive` $\leftarrow$ `shape_in_triangle`$(Z)$. The precision and recall of the rule are both 1.

Specifically, the LLMs used for translating semantics include Gemini 2.5 Pro, GPT-5, and GPT-o3. They output the same semantics for one predicate placeholder based on Table 5 in Appendix D, which indicates that the learned semantics of predicates by $\gamma$ILP are easy to be easily translated by the current LLMs.

## 6 CONCLUSION

In this work, we presented the fully differentiable rule-based inductive learning pipeline $\gamma$ILP for relational images and pure images without the symbolic labels of image constants. Firstly, $\gamma$ILP transforms the symbol grounding process by employing encoders and a clustering module to assign representations to image constants. Secondly, the differentiable ground substitution facilitates first-order rule learning with GPUs. Thirdly, it tackles predicate invention through interpreting the constant represented by the variables in the learned first-order rules. For the experimental evaluation, we considered classical ILP datasets, relational image datasets, and pure image datasets such as Kandinsky patterns. The results show that $\gamma$ILP effectively learns first-order logic rules, achieves strong classification performance, and successfully induces predicate semantics. For future work, we believe learning rules to explain the image with spatial information (Zhang et al., 2019), introducing simple language bias for learn longer rules, and considering the multimodal inputs are promising.

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

# A  STATISTICAL INFORMATION OF ILP DATASETS

To assess our differentiable substitution method, we evaluate $\gamma$ILP on classical ILP datasets (Evans & Grefenstette, 2018). Constants are textual; thus, we set $g(\mathbf{x}) = \mathbf{x}$ and $\lambda = 0$ in Eq. (4). We use pre-trained VAE as the encoder for textual relations and constants. We report results from baseline models, including $\partial$ILP (Evans & Grefenstette, 2018), DFORL (Gao et al., 2024), Gemini 2.5 Pro, and GPT-5. Each experiment was run ten times under different random seeds. The maximum running time for $\gamma$ILP is set to 5 minutes. We report the average recall of the learned rules with precision equal to 1.

For $\partial$ILP, we calculated the recall based on the rules reported in their paper. To eliminate predicate semantics leakage, we replaced the same predicate with a consistent random string across runs for all models. The number of constants and relations in the training set under each task of the classical inductive logic programming (ILP) datasets is shown in Table 3. When testing data is available, such as in the Husband and Uncle tasks, we compute the recall of the learned rules on the test set. Otherwise, recall is computed on facts involving constants not seen during training.

| Domain | Task | # Constant | # Relation |
|---|---|---|---|
| Arithmetic | Predecessor | 10 | 3 |
| | Odd | 10 | 3 |
| | Even | 10 | 3 |
| | Lessthan | 10 | 3 |
| | Fizz | 7 | 3 |
| | Buzz | 10 | 3 |
| Lists | Member | 8 | 3 |
| | Length | 8 | 3 |
| Family Tree | Son | 9 | 4 |
| | Grandparent | 9 | 3 |
| | Husband | 2102 | 12 |
| | Uncle | 2102 | 12 |
| | Relatedness | 8 | 2 |
| | Father | 6 | 5 |
| Graphs | Undirected Edge | 4 | 2 |
| | Adjacent to Red | 7 | 5 |
| | Two Children | 5 | 2 |
| | Graph Coloring | 8 | 3 |
| | Connectedness | 4 | 2 |
| | Cyclic | 6 | 3 |

Table 3: The statistical information of ILP datasets.

The expected rules learned by $\gamma$ILP in classical ILP datasets are the same as (Gao et al., 2024). The recall comparison results on the ILP datasets are shown in Table 4. $\gamma$ILP finds expected rules under classical ILP tasks except for fizz and buzz datasets, matching the performance of DFORL. In the Fizz and Buzz datasets, the rules with 1 recall contain four and six variables, respectively. Consequently, the search space for $\gamma$ILP and DFORL becomes enormous without constraints such as the logical template used in $\partial$ILP. The rules learned by $\gamma$ILP in Fizz and Buzz tasks are presented as follows:

$$\mathtt{fizz}(X) \leftarrow \mathtt{zero}(X). \tag{5}$$
$$\mathtt{fizz}(X) \leftarrow \mathtt{succ}(Z, Y), \mathtt{fizz}(Y), \mathtt{succ}(Z, X). \tag{6}$$
$$\mathtt{buzz}(X) \leftarrow \mathtt{zero}(X). \tag{7}$$
$$\mathtt{buzz}(X) \leftarrow \mathtt{succ}(Z, Y), \mathtt{buzz}(Y), \mathtt{succ}(Z, X). \tag{8}$$

In $\gamma$ILP, the learned rule can only describe the constant zero, but fails to capture numbers divisible by three and five in the Fizz and Buzz datasets, respectively. For the rules reported in $\partial$ILP cover the most tasks except the Husband task, where the rules are incomplete. When using all relational facts as inputs, Gmini 2.5 Pro can also induce all the expected logic rules in all tasks. However, without

| Domain | Task | ∂ILP | DFORL | GPT-5 | Gemini | γILP | RT |
|--------|------|------|-------|-------|--------|------|-----|
| Arithmetic | Pre | 1.00 | 1.00 | 1.00 | 1.00 | 1.00 | 6.57 |
| | Odd | 1.00 | 1.00 | 1.00 | 1.00 | 1.00 | 6.98 |
| | Even | 1.00 | 1.00 | 1.00 | 1.00 | 1.00 | 6.47 |
| | Lessthan | 1.00 | 1.00 | 1.00 | 1.00 | 1.00 | 12.09 |
| | Fizz | **1.00** | - | **1.00** | **1.00** | - | 35.55 |
| | Buzz | **1.00** | - | **1.00** | **1.00** | - | 133.03 |
| Lists | Member | 1.00 | 1.00 | 1.00 | 1.00 | 1.00 | 5.50 |
| | Length | 1.00 | 1.00 | 1.00 | 1.00 | 1.00 | 12.08 |
| Family | Son | 1.00 | 1.00 | 1.00 | 1.00 | 1.00 | 7.77 |
| Tree | GP | 1.00 | 1.00 | 1.00 | 1.00 | 1.00 | 61.09 |
| | Husband | 0.50 | **1.00** | 0.00 | **1.00** | **1.00** | 42.39 |
| | Uncle | **1.00** | **1.00** | 0.00 | **1.00** | **1.00** | 76.50 |
| | Rel | 1.00 | 1.00 | 1.00 | 1.00 | 1.00 | 46.94 |
| | Father | 1.00 | 1.00 | 1.00 | 1.00 | 1.00 | 5.46 |
| Graphs | UE | 1.00 | 1.00 | 1.00 | 1.00 | 1.00 | 8.48 |
| | AdjR | 1.00 | 1.00 | 1.00 | 1.00 | 1.00 | 7.39 |
| | TC | 1.00 | 1.00 | 1.00 | 1.00 | 1.00 | 5.82 |
| | GC | 1.00 | 1.00 | 1.00 | 1.00 | 1.00 | 17.35 |
| | Con | 1.00 | 1.00 | 1.00 | 1.00 | 1.00 | 5.82 |
| | Cyclic | 1.00 | 1.00 | 1.00 | 1.00 | 1.00 | 20.66 |

Table 4: The recall of the learned rules on the ILP datasets. For GPT and Gemini, we use the latest reasoning model GPT-5 and Gemini 2.5 Pro, respectively. "Pre", "GP", "Rel", "UE", "AdjR", "TC", "GC", "Con", and "RT" denote Predecessor, Grandparent, Relatedness, Undirected Edge, Adjacent to Red, Two Children, Graph Coloring, Connectedness, and the average running time in second for γILP when it reaches its maximum accuracy, respectively. The best results are in bold if some models fail to achieve the top recall. The notation "–" indicates that the model achieves a recall between 0 and 1.

any data preprocessing, GPT-5 cannot learn from huge datasets with limited scalability under the Husband and Uncle datasets. However, when we only select the first 50 relational facts as input, GPT-5 can also induce the correct rules under both the Husband and Uncle datasets.

## B   RULES LEARNED BY LLMS FROM RELATIONAL IMAGE DATASETS

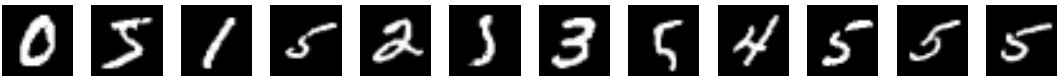

Figure 4: The MNIST sequence with label $0, 5, 1, 5, 2, 5, 3, 5, 4, 5, 5, 5 \ldots$

When learning from relational image datasets, we replace each predicate with a unique random string and annotate each image with a random identifier. Then, we replace each constant in the classical ILP benchmarks with the annotation of an image whose label matches the constant's value. Next, we use a fixed-format prompt to induce logic programs using LLMs. For example, in the Fizz task, we have the fact: zero(0, 0), succ(0, 1), succ(1, 2), succ(2, 3), succ(3, 4), succ(4, 5), succ(5, 6), fizz(0, 0), fizz(3, 3), fizz(6, 6). Note that for atoms with unary predicates, we rewrite them by duplicating the constant to form a binary structure. We replace the zero predicate with "4WY", the succ predicate with "7h0", and the fizz predicate with "xoh". We then randomly select MNIST images and assign each one a random annotation. Then, the formatted prompt for LLMs is: "If you have the following images and their annotations, you also have the fact set in $r(a_1, a_2)$, where $r$ indicates the relation between the image annotated with $a_1$ and the image annotated with $a_2$. All facts are: 4WY(KI5, fRB), 7hO(t01, 1kn), 7hO(Yjp, NRS), 7hO(ySZ, 4bI), 7hO(bL1, qfI), 7hO(4md, VY4), 7hO(qOg, IdR), xoh(4Qu, kUw), xoh(uNT, 3HN), xoh(99c, Fwv). Can you learn a first-order logic program to describe the "xoh" relation with existing relations and images?" Then, we present the rules learned by Gemini 2.5 Pro and GPT-o3 in the relational MNIST image datasets.

**Even Task.** The logic program $P$ learned by Gemini 2.5 Pro for the Even task is:

$$\text{even}(A) \leftarrow \text{succ}(A,B), \text{succ}(B,C).$$

The rule learned by Gemini 2.5 Pro is not expected because the precision is not 1.

However, GPT-o3 can learn the expected rules for the Even task.

$$\text{even}(X) \leftarrow \text{zero}(X).$$
$$\text{even}(C) \leftarrow \text{succ}(A,B), \text{succ}(B,C), \text{even}(A)$$

**Odd Task.** When we use Gemini 2.5 Pro to learn odd predicates in the Odd task. The learned expected rules are:

$$\text{odd}(A) \leftarrow \text{succ}(B,A), \text{zero}(B).$$
$$\text{odd}(A) \leftarrow \text{succ}(B,A), \text{succ}(C,B), \text{zero}(C).$$

However, GPT-o3 fails to learn any generalized rules to describe the Odd task, and it only outputs the facts:

$$\text{odd}(W8h).\text{odd}(Pay).\text{odd}(UhR).\text{odd}(8km).\text{odd}(7Oh).$$

In the above, W8h, Pay, UhR, 8km, and 7Oh are the annotations corresponding to the images with labels 1, 3, 5, 7, 9, respectively. In addition, these output facts are also already occurring in the prompt.

When we indicate the predicate name for each relation annotation, GPT-o3 can learn the expected rules for the Odd task:

$$\text{odd}(X) \leftarrow \text{succ}(Z,X), \text{zero}(Z)$$
$$\text{odd}(X) \leftarrow \text{succ}(Z,X), \text{succ}(Y,Z), \text{odd}(Y)$$

**Predecessor Task.** When we use Gemini 2.5 Pro to learn the predecessor predicate in the Predecessor task, the learned rules are:

$$\text{pre}(A,B) \leftarrow \text{digit}(A,\text{Val1}), \text{digit}(B,\text{Val2}),$$
$$\text{Val1 is Val2} + 1.$$

Considering the generalization ability of LLMs, we believe this rule is correct, as it can generate all relevant examples despite lacking the formalized predicates observed during training.

For GPT-o3, the learned rule is also correct:

$$\text{pre}(X,Y) \leftarrow \text{succ}(Y,X).$$

**Lessthan Task.** When we use Gemini 2.5 Pro to learn the `lessthan` predicate in the Lessthan task, the learned rule is:

$$\text{lessthan}(X,Y) \leftarrow \text{ is digit}(X,Dx),$$
$$\text{is digit}(Y,Dy), Dy \text{ is } Dx + 1,$$

which is marked as incomplete.

For GPT-o3, the model learns the following correct rules for the Lessthan task, demonstrating the generalization ability of LLMs:

$$\text{same}(A,B) \leftarrow A = B.$$
$$\text{same}(A,B) \leftarrow \text{zero}(A,B).$$
$$\text{bigger}(A,B) \leftarrow \text{succ}(A,B).$$
$$\text{bigger}(A,B) \leftarrow \text{succ}(A,C), \text{bigger}(C,B).$$
$$\text{lessthan}(X,Y) \leftarrow \text{same}(X,X1),$$
$$\text{same}(Y,Y1),$$
$$\text{bigger}(X1,Y1).$$

**Fizz Task.** We also use Gemini 2.5 Pro to learn the Fizz predicate in the Fizz task. The learned rules are incorrect, where "not" is negation:

$$\texttt{fizz}(A) \leftarrow \text{not } \texttt{relation\_participant}(A).$$
$$\texttt{relation\_participant}(A) \leftarrow \texttt{succ}(A, \_).$$
$$\texttt{relation\_participant}(A) \leftarrow \texttt{succ}(\_, A).$$
$$\texttt{relation\_participant}(A) \leftarrow \texttt{zero}(A).$$

In addition, GPT-o3 learns an incorrect rule to describe the Fizz task:

$$\texttt{fizz}(A) \leftarrow \texttt{succ}(A, \_), \texttt{succ}(\_, A), \texttt{zero}(A),$$

where the placeholder _ indicates any constants.

**Buzz Task.** When using Gemini 2.5 Pro to learn the Buzz predicate in the Buzz task, the resulting rule is incorrect:

$$\texttt{Reg}(A) \leftarrow \text{not } \texttt{zero}(A), \text{not } \texttt{succ}(A, \_), \text{not } \texttt{succ}(\_, A).$$

where _ is a placeholder for any image annotation.

However, the output by GPT-o3 for the Buzz task is correct:

$$\texttt{plus5}(X, Y) \leftarrow \texttt{succ}(X, A1), \texttt{succ}(A1, A2),$$
$$\texttt{succ}(A2, A3), \texttt{succ}(A3, A4),$$
$$\texttt{succ}(A4, Y).$$
$$\texttt{buzz}(X) \leftarrow \texttt{plus5}(Y, X), \texttt{zero}(Y).$$
$$\texttt{buzz}(X) \leftarrow \texttt{zero}(X).$$

## C  MORE $\gamma$ILP-GENERATED RULES FROM THE TWO-PAIR TASK

In this section, more rules for describing the two-pair task in Kandinsky patterns generated by $\gamma$ILP are presented as follows.

$$\texttt{positive} \leftarrow p(Y, R) \tag{9}$$
$$\texttt{positive} \leftarrow p_1(X), p_2(Y) \tag{10}$$

In these rules, the constants represented by the variables $R$, $X$, and $Y$ are presented in Figure 5. By querying the semantics of the predicate placeholders using LLMs, we obtain the following interpretations: $p$ indicates "different color, shape in triangle", $p_1$ indicates "shape in triangle and color in yellow", and $p_2$ indicates "shape in triangle and color in red". These rules only capture the pair with the same shape but different colors, while the other required pair, with the same shape and color, is not covered by the rule. Hence, the precision of the rule in (9) and the rule in (10) is 0.5 and 0.75, respectively. The recall values of the two specific rules are less than 1.

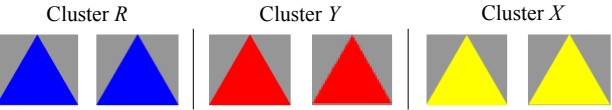

Figure 5: The image constants represented by the clusters $R$, $X$, and $Y$.

In addition, we also generate the following rule from the two-pair task:

$$\texttt{positive} \leftarrow p_1(Y), p_2(Z), p_3(T). \tag{11}$$

For this rule, the constants represented by the clusters $T$, $Y$, and $Z$ are presented in Figure 6. This rule covers that the constants in one pair have the same color but different shapes (see the constants under the clusters $T$ and $Y$), and the constants in another pair have the same color and shape (see the image constants under the cluster $Z$). The precision of this rule is 1, which ensures that the images are classified without any false positive instances. However, the recall of this rule is less than 1, as it captures only one specific pattern combination in the two-pair task.

Figure 6: The constants represented by the clusters $T$, $Y$, and $Z$.

| Tasks | GPT-5 | GPT-3o | Gemini 2.5 Pro |
|---|---|---|---|
| Two-pair | Cluster R and Cluster V represent the same shape category, but differ in color. | Everything about the paired images is identical—shape, size, cropping, background—except for the hue of the triangle. | The relationship is one of contrast in the color dimension, while maintaining consistency in the shape and overall format. |
| One-red | They are all solid red geometric shapes. | Every item in that set is filled with the same solid red colour; the hue is the property they all share. | color (red) is the property they all possess in an identical state, making it the primary commonality. |
| One-triangle | All the objects in the image share one clear common property: They are all triangles. | Every one of the thumbnails shows the same basic geometric shape: an upright, isosceles (appearance-wise almost equilateral) triangle. Color, slight aliasing and exact position vary, but the common property is simply that they are all triangles pointing upward inside the square frame. | Based on the provided image, the most immediate and common property of all the distinct triangular objects shown is that they are all isosceles triangles. |

Table 5: The semantics translation results from LLMs

## D  LLMS AS TRANSLATORS

We test the translated semantics of predicate placeholders in natural language by using LLMs such as GPT-5, GPT-3o, and Gemini 2.5 Pro in Table 5. We use the above LLMs on their official user interface. The translated results are the same for GPT5, GPT-3o, and Gemini 2.5 Pro, which indicates the semantics of the predicate placeholder learned by $\gamma$ILP are not sensitive to different LLMs.

## E  EXPLANATIONS OBTAINED BY LLMS AND OTHER LEARNING MODELS ON KANDINSKY PATTERNS

This section evaluates predictions from reasoning-capable LLMs (Gemini 2.5 Pro, GPT-o3, GPT-5) alongside hybrid methods that pair clustering with RIPPER or C4.5.

When there are no well-defined relationships in the instance, we also use RIPPER (Cohen, 1995) and C4.5 with ViT as encoder (Quinlan, 1993) to replace the first-order rule learning module in $\gamma$ILP. They classify the instance classes based on the centroid indices of image constants in the instances. The accuracies are presented in Table 6. The learned propositional rules have less interpretability compared with the rules learned by $\gamma$ILP.

**Rules generated from Gmini 2.5 Pro.**  The inductive explanation from Gemini 2.5 Pro for the two-pair task is: "The number of constants of each shape type is an even number (i.e., 2 or 4 of

| Task | RIPPER-ViT | C4.5-ViT | $\gamma$ILP-VAE | $\gamma$ILP-ViT |
|------|-----------|----------|-----------|-----------|
| Types | RM | RM | RM | RM |
| Two-pair | 0.70 | 0.65 | 0.64 | **0.75** |
| One-red | **1.00** | **1.00** | 0.77 | **1.00** |
| One-triangle | **1.00** | **1.00** | 0.77 | **1.00** |

Table 6: Classification accuracy on Kandinsky tasks: Two-pair, one-red, and one-triangle.

each shape present), and the image contains constants of at least two different colors." However, this explanation is incorrect with very low precision. In contrast, for the one-red and one-triangle tasks, the explanations are correct and complete. These results suggest that while the state-of-the-art reasoning model Gemini 2.5 Pro performs well on simpler reasoning tasks, it struggles with more complex tasks, such as two-pair, which require analyzing the composition of multiple constants. In such cases, the model fails to generate correct inductive explanations.

**Rules generated from GPT-5.** For GPT-5, the latest reasoning model from OpenAI, the explanation for the one-red task is: "The rule is simple — a panel is true if it contains any red shape." For the one-triangle task, the learned rule is: "An image is labeled true if it contains at least one triangle; it's labeled false if it contains no triangles." For the two-pair task, the induced rule by GPT-5 is: "An image is true if it contains an even number of triangles $(0, 2, 4, \dots)$. It's false if the number of triangles is odd." We conclude that GPT-5 can induce correct and complete rules in one-red and one-triangle tasks. However, the ability to use this knowledge directly and classify the test images is not mature enough now (see Table 2). In addition, for more complex tasks, such as two-pair, the rule induced by GPT-5 has low precision and low recall. Hence, GPT-5 cannot classify two-pair Kandinsky patterns with 100% accuracy.

**Rules generated from GPT-o3.** For GPT-o3, the model generates the following explanations for the two-pair task: "Positive = all foreground shapes share exactly one colour. Negative = two or more different foreground colours appear." However, this explanation has precision 0 and recall 0. For the one-red task, it outputs: "Positive examples satisfy the 'all-four-shapes-same-size' condition; negatives include any other case (different sizes and/or a count different from 4)." This rule also has low precision and recall. For the one-triangle task, the model explains: "An image is positive when it contains the same number of circles and squares; otherwise, it is negative." This explanation also has low precision and recall.

**Rules generated from RIPPER.** For the clustering method with RIPPER, the learned rule under the one-red task is:

$$\text{positive} \leftarrow \#6. \tag{12}$$
$$\text{positive} \leftarrow \#7. \tag{13}$$
$$\text{positive} \leftarrow \#3. \tag{14}$$

where each body atom represents a cluster centroid index. The image constants represented by these centroids are presented in Figure 7. We can also infer that the red color, no matter the shape, is key to the information to determine the classes of the one-red pattern images.

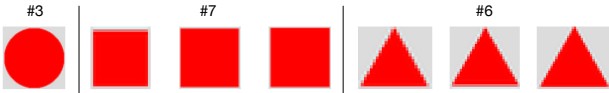

Figure 7: The constants represented by the cluster centroids #3, #6, and #7.

The learned rules under the one-triangle task are:

$$\text{positive} \leftarrow \#7. \tag{15}$$
$$\text{positive} \leftarrow \#6. \tag{16}$$

where the constants represented by the centroids are presented in Figure 8. Hence, the shape of a triangle, regardless of its color, is the key information to determine the classes of one-triangle pattern images.

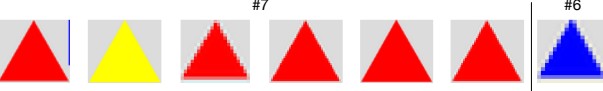

Figure 8: The constants represented by the cluster centroids #6 and #7.

In the two-pair task, RIPPER generates two rules:

$$\texttt{positive} \leftarrow \text{not } \#8 \wedge \text{not } \#6 \wedge \text{not } \#0. \tag{17}$$
$$\texttt{positive} \leftarrow \text{not } \#1 \wedge \text{not } \#7 \wedge \#8. \tag{18}$$

The image constants represented by the body atoms in rule (17) are presented in Figure 9a and the image constants represented by the body atoms in rule (18) are presented in Figure 9b. However, based on these two rules, we cannot induce any possible predicates based on the rules in the propositional format, nor explain the two-pair pattern.

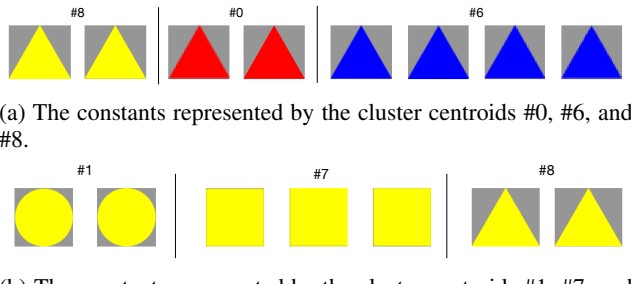

(a) The constants represented by the cluster centroids #0, #6, and #8.

(b) The constants represented by the cluster centroids #1, #7, and #8.

Figure 9: The constants represented by the cluster centroids #0, #1, #6, #7, and #8.

**Rules generated from C4.5.** We use the same clustering assignment for running C4.5 and RIPPER, the rules under the one-red task are the same as rules (12) to (14), and the constants represented by the body atoms are presented in Figure 7. The rules under the one-triangle task are the same as the rules (15) to (16), and the constants represented by the cluster centroids are also presented in Figure 8. For the two-pair task, the learned rule is:

$$\texttt{positive} \leftarrow \text{not } \#1 \wedge \text{not } \#7 \wedge \text{not } \#8 \wedge \text{not } \#6 \wedge \#2,$$

where the cluster centroids are presented in Figure 10. Given this rule, we are unable to induce the knowledge required to classify two-pair patterns.

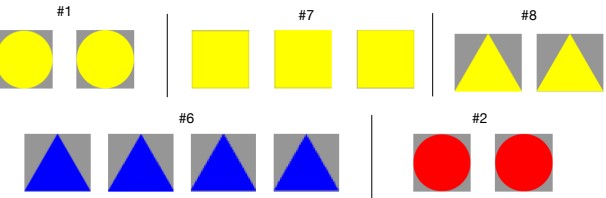

Figure 10: The constants represented by the cluster centroids #1, #2, #6, #7, and #8.

# F  ABLATION AND ANALYSIS

In this section, we test the sensitivity of our model on Kandinsky patterns to various hyperparameters, including the number of clusters, the learning rate of the rule learning network, the learning

rate for the differentiable clustering method, the hyperparameter $\alpha$ defined in the differentiable clustering method described by Eq. (2), and the hyperparameter $\gamma$ described in Eq. (4). In this setting, we increase the training instances in both the training and testing datasets. Due to the total instance number in each Kandinsky pattern task being 100, we set 80 and 20 instances to the training dataset and the testing dataset, with balanced positive and negative instances, respectively. For the one-red and one-triangle tasks, the base parameter values for the number of clusters, learning rate of the rule learning network, the learning rate of the differentiable clustering method, $\alpha$, and $\gamma$ are 10, 0.05, 0.5, 20, and 4, respectively. For the two-pair tasks, the base parameter values for the number of clusters, learning rate of the rule learning network, the learning rate of the differentiable clustering method, $\alpha$, and $\gamma$ are 10, 0.5, 0.1, 20, and 4, respectively. We compare accuracy across different values of a single hyperparameter while keeping all other hyperparameters fixed. We ran each experiment five times under each setting and recorded the best results for each setting. The accuracies are presented in Figure 11. The experimental results show that the differentiable cluster learning rate and $\alpha$ have a smaller impact on accuracy compared with the number of clusters, the learning rate of the rule learning network, and $\lambda$. Moreover, adjusting the centroids during training (when $\lambda > 0$) leads to higher accuracy than training with fixed centroids (when $\lambda = 0$) on both the one-red and two-pair tasks.

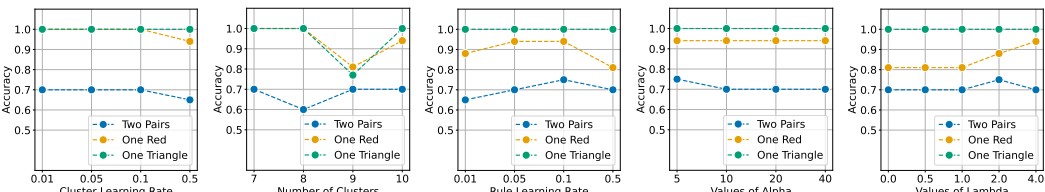

(a) Different DCM LR (b) Different cluster counts. (c) Different rule learning network LR. (d) Different $\alpha$ values. (e) Different $\lambda$ values.

Figure 11: The accuracies under different hyperparameters. DCM and LR indicate the differentiable clustering method and learning rate, respectively.

Furthermore, we analyze the $\gamma$ILP's stability on accuracy in Kandinsky patterns. We collect accuracies under 10 runs with different seeds. For the one-red and one-triangle tasks, the cluster number is 8, the rule learning rate is 0.05, the differentiable clustering method learning rate is 0.5, the value of $\alpha$ is 20, and the value of $\lambda$ is set to 4. For the two-pair task, the cluster number is 10, the rule learning rate is 0.5, the differentiable clustering method learning rate is 0.1, the value of $\alpha$ is 5, and the value of $\lambda$ is set to 4. The stability results are presented in Figure 12.

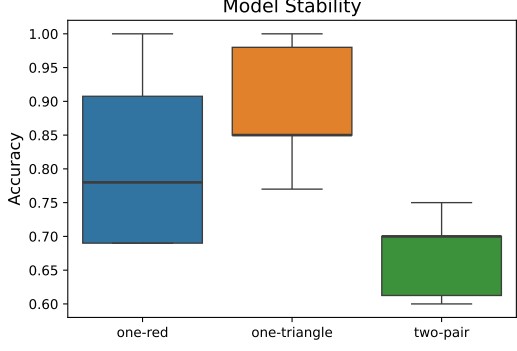

Figure 12: The stability of $\gamma$ILP in Kandinsky patterns.

# G   LLM USAGE

The LLMs are used as a baseline and a tool for predicate invention in the paper. Additionally, LLMs are utilized to aid or polish the writing.

