# OpenReview forum: "Visual Perceptual to Conceptual First-Order Rule Learning Networks"
_ICLR.cc/2026/Conference — Submitted to ICLR 2026_

### Official Review · Reviewer_AuPA · 2025-10-25

**Soundness:** 3
**Presentation:** 3
**Contribution:** 3
**Rating:** 6
**Confidence:** 3

**Summary:**

This paper tackles the problem of learning explainable, first-order logical rules directly from raw images without relying on any image labels. The proposed method, γILP, is a differentiable framework that trains a deep clustering module and a differentiable logic learner, forcing the system to discover concepts that are both visually and logically consistent. The main results show that γILP can successfully learn rules from unlabeled images competitive with or better than certain LLMs, and can match the accuracy of black-box classifiers (like ResNet) on abstract tasks while also inventing the correct, human-readable rules for its decisions.

**Strengths:**

Learning interpretable rules directly from perceptual data is a hard problem and this paper proposes a valid solution for it.

Interpretebility has been a limitation for deep learning methods, this paper addresses this by combining symbolic approaches with deep learning to learn interpretable rules.

The interplay between symbolic and deep learning based methods is well done and leads to human interpretable rules.

**Weaknesses:**

I believe one limitation is its not clear to me how this approach can scale to real-world images where
1. Clustering may not work well
2. rules may not be very clear and interpretable

Also, the number of cluster should be known beforehand which seems like a limitation

**Questions:**

A clarification on the LLM baseline experiments How many example images from a latent cluster were typically fed into the LLM? Given that a large number of images could exhaust the context window, how was this image sampling handled? Did the authors investigate the sensitivity of the LLM to this?

---

> ### Author Response · Authors · 2025-11-21
> **Response to Questions and Weaknesses**
>
> Thank you for your support of our rule learning pipeline. Please see the detailed response below.
>
> **Questions**
>
> >A clarification on the LLM baseline experiments: How many example images from a latent cluster were typically fed into the LLM? Given that a large number of images could exhaust the context window, how was this image sampling handled?
>
> We understand the concern regarding the sensitivity of using LLMs to translate predicate semantics. In our experiments, we randomly selected 20 constants from the full set of constants in each Kandinsky pattern dataset. Within each cluster, the number of images ranges from 3 to 8, and examples of the images fed into the model are shown in Figures 2(d) and 3. All constants within each cluster were provided to the LLMs without any human selection or filtering.
>
>
>
> >Did the authors investigate the
> sensitivity of the LLM to this?
>
> We maintain that predicate invention is performed by $\gamma$ILP based on the learned structural rules, the ordering of variable clusters, and the constants within each cluster.
> LLMs are used only to automatically translate the learned semantics.
> Since the rules themselves do not depend on the LLMs, we design a simple and explicit prompt solely to translate the semantics. Therefore, prompt sensitivity is not within the scope of this setting.
>
> We additionally collect the outputs of the semantic translations generated by GPT-5, GPT-3o, and Gemini 2.5 Pro.
> For the two-pair task, we use the prompt: ‘What is the relation between images in cluster R and images in cluster V?’ and evaluate it using the images in clusters $R$ and $V$ shown in Figure 2(d).
> For the one-red task, we use the prompt: ‘What is the common property of the objects in this image?’ and test it with the image in cluster $U$ in Figure 3(a).
> For the one-triangle task, we use the same prompt and evaluate it with the image shown in Figure 3(b). The results are presented as:
>
> | Tasks | GPT-5 | GPT-3o | Gemini 2.5 Pro |
> | --- | --- | --- | --- |
> | Two-pair | Cluster R and Cluster V represent the same shape category, but differ in color. | Everything about the paired images is identical—shape, size, cropping, background—except for the hue of the triangle. | The relationship is one of contrast in the color dimension, while maintaining consistency in the shape and overall format. |
> | One-red | They are all solid red geometric shapes. | Every item in that set is filled with the same solid red colour; the hue is the property they all share. | color (red) is the property they all possess in an identical state, making it the primary commonality. |
> | One-triangle  | All the objects in the image share one clear common property: They are all triangles. | Every one of the thumbnails shows the same basic geometric shape: an upright, isosceles (appearance-wise almost equilateral) triangle. Color, slight aliasing and exact position vary, but the common property is simply that they are all triangles pointing upward inside the square frame. | Based on the provided image, the most immediate and common property of all the distinct triangular objects shown is that they are all isosceles triangles. |
>
> The experimental results indicate that the semantics can be readily translated by all the LLMs considered. We also append these further experiment results in Table 5 of Appendix D.
>
>
> **Weaknesses**
>
>
> >I believe one limitation is its not clear to me how this approach can scale to real-world images where 1. Clustering may not work well
>
> We agree that clustering quality is task-dependent. Our experimental setup focuses on tasks where the underlying structure is well suited for cluster-based predicate invention. We will further evaluate the model’s performance on additional real-world datasets and explore different self-supervised methods.
>
> >rules may not be very clear and interpretable
>
> Theoretically, a (datalog-like) first-order language provides stronger expressive power for describing the facts hidden in the data than propositional logic rules.
> $\gamma$ILP aims to describe images with or without predefined relations.
> To interpret the images, the learned rules contain predicate placeholders. We believe these predicate placeholders accurately capture the underlying concepts between constants. Moreover, the variables naturally refer to the constants in realistic settings.
> Therefore, we maintain that the resulting rules are easy to interpret.
>
> >Also, the number of cluster should be known beforehand which seems like a limitation
>
> The number of clusters is a hyperparameter. In practice, we set it to a relatively large value to ensure sufficient capacity for rule generation. We also analyze the sensitivity of this hyperparameter as part of our ablation study, as shown in Figure 11 in Appendix E.

---

### Official Review · Reviewer_dz7p · 2025-10-29

**Soundness:** 2
**Presentation:** 3
**Contribution:** 2
**Rating:** 4
**Confidence:** 3

**Summary:**

The paper proposes a neuro‑symbolic framework for learning first‑order logic rules directly from images without explicit labels. The pipeline couples (1) a perceptual front‑end (pre‑trained encoders), (2) a differentiable clustering module that induces latent “symbolic” entities, (3) a differentiable substitution mechanism that builds training examples for (4) a differentiable rule‑learning network, and (5) an LLM‑based predicate invention step that maps placeholder predicates to human‑readable semantics. Experiments span classical ILP (Inductive logic programming) tasks, relational MNIST variants, and Kandinsky patterns. The system shows encouraging results on visual reasoning benchmarks, with qualitative examples of invented predicates.

**Strengths:**

- A coherent architecture that connects perception, clustering as latent symbol induction, and differentiable rule learning, culminating in interpretable rules. The overall design targets a long‑standing neuro‑symbolic goal—bridging pixels to logic.

- On compositional visual reasoning (e.g., two‑pair), the method outperforms standard vision baselines and illustrates the value of explicit relational structure.

- The paper is well‑structured, with helpful figures/tables and algorithm boxes that make the pipeline understandable to readers outside ILP.

**Weaknesses:**

- I believe paper's central claim of a "fully differentiable" pipeline relies on a non-differentiable. The predicate invention step, as detailed in Algorithm 2, relies on a non-differentiable, hard query to a frozen LLM. This is the step where the actual "conceptual" label (e.g., "triangle") is introduced, and it lies completely outside the end-to-end optimization loop. This architectural choice has implications -- the model does not learn concepts like "red" or "triangle" through backpropagation. Instead, it learns to form a cluster of entities that happen to be red triangles, and then it asks an external oracle (the LLM) for the appropriate human-readable label. This should be stated explicitly and transparently if my interpretation is correct as it reframes the contribution.

- The reported runtimes (Table 4, Appendix) focus on γILP's own training steps but do not discuss the immense computational resources required by ViT, BERT, and especially LLM queries. Training and inferring with these foundation models comes with high cost in time, compute, and financial resources. There is no holistic accounting of these requirements, nor is the LLM inference cost (token count, latency, etc.) analyzed.

- Some experiments appear to encode label cardinality via the number of clusters and rely on pre‑trained encoders with class‑discriminative structure, blurring the “no label leakage” claim.

- Failures on classical tasks (Fizz/Buzz) signal a combinatorial bottleneck in bottom‑up propositionalization with many variables/longer chains.

- Several results are reported as “best of N runs,” with substantial variance on harder tasks. It will be helpful to see mean±std over fixed seeds and confidence intervals.

**Questions:**

- How does semantic accuracy change with 10–20% noisy cluster members? Please include a prompt‑sensitivity table across 3–4 prompt families and 2–3 LLMs if you have any.

- What concrete search constraints or hierarchical strategies can unlock 4+ variable tasks (Fizz/Buzz)?

- What is the end‑to‑end costing (encoder inference + LLM queries): tokens per predicate, latency, and monetary cost per dataset.

- What do you think of learnable truth‑scoring for placeholder atoms instead of a strict logical AND over entity presence?

---

> ### Author Response · Authors · 2025-11-21
> **Response to Questions 1 and 2**
>
> **Questions**
> >How does semantic accuracy change with 10–20\% noisy cluster members? Please include a prompt‑sensitivity table across 3–4 prompt families and 2–3 LLMs if you have any.
>
> We believe that you are concerned about whether the prompt or LLMs are sensitive to the query.
>
> To automatically obtain the semantics, $\gamma$ILP outputs a rule containing a predicate placeholder, the ordered variables, and the constant images represented by these variables. This output inherently captures the semantics of the predicate placeholder by analyzing the image constants. In fact, the output of $\gamma$ILP reflects the interpretability of the entire framework.
>
> Regarding the prompt, we believe that the rules learned by $\gamma$ILP already capture the semantics of the predicate placeholders. Therefore, we use a single, explicit, and simple query with the constants represented by variables. We randomly selected 20 constants from all the constants in the background knowledge. In this setup, the LLMs serve solely as translators, converting the semantics learned by $\gamma$ILP into natural language.
>
> Rather than adding 10–20\% noise to evaluate interpretability, we care more about the precision and recall of rules learn by $\gamma$ILP. Based on these two indicators, we can get the precise semantics of the predicate placeholder.
>
>
> Based on your concern, we chose GPT-5, GPT-3o, and Gemini 2.5 Pro to compare the translation ability from the output of $\gamma$ILP. For the two-pair task, we use the prompt:
> 'what is the relation between images under cluster R and the images under cluster V?' and test with the images under clusters $R$ and $V$ Figure 2 (d).
> For the one-red task, we use the prompt: 'What is the common property of these objects in the image?' and test with the image under the cluster $U$ in Figure 3 (a). For the one-triangle task, we use the prompt:' What is the common property of these objects in the image?' and test with the image in Figure 3 (b).  The results are presented as follows:
> | Tasks | GPT-5 | GPT-3o | Gemini 2.5 Pro |
> | --- | --- | --- | --- |
> | Two-pair | Cluster R and Cluster V represent the same shape category, but differ in color. | Everything about the paired images is identical—shape, size, cropping, background—except for the hue of the triangle. | The relationship is one of contrast in the color dimension, while maintaining consistency in the shape and overall format. |
> | One-red | They are all solid red geometric shapes. | Every item in that set is filled with the same solid red colour; the hue is the property they all share. | color (red) is the property they all possess in an identical state, making it the primary commonality. |
> | One-triangle  | All the objects in the image share one clear common property: They are all triangles. | Every one of the thumbnails shows the same basic geometric shape: an upright, isosceles (appearance-wise almost equilateral) triangle. Color, slight aliasing and exact position vary, but the common property is simply that they are all triangles pointing upward inside the square frame. | Based on the provided image, the most immediate and common property of all the distinct triangular objects shown is that they are all isosceles triangles. |
>
> The experimental results indicate that the semantics can be easily inferred by all the considered LLMs without inconsistency.
>
>
> >What concrete search constraints or hierarchical strategies can unlock 4+ variable tasks (Fizz/Buzz)?
>
> For our setting, we use the learning form bottom-up inductive logic programming method to learning rules without any  constraints or language bias.
> We believe that introducing structural constraints can further improve scalability. In fact, we are currently investigating some options, such as imposing a constraint that enforces a tree-shaped structure on rule bodies. This idea is inspired by existential query answering, where acyclicity in the query ensures controlled complexity (see ``Algorithms for acyclic database schemes'' of Yannakakis [1]).
> Besides, [2] introduces a  logic template, which is a language bias that speculates the order of variables in the learned rules.
> We believe such constraints could significantly reduce the search space in our setting as well.
> That said, this extension involves nontrivial design choices and additional theoretical discussion, and we view it as an important direction for future work rather than something that can be included in the present submission due to space limitations.
>
>
>
>
> Reference:
>
> [1] Mihalis Yannakakis: Algorithms for Acyclic Database Schemes. VLDB 1981: 82-94.
>
> [2] Evans R, Grefenstette E. Learning explanatory rules from noisy data. Journal of Artificial Intelligence Research. 2018 Jan 26;61:1-64.

---

> ### Author Response · Authors · 2025-11-21
> **Response to the Remaining Questions and Weaknesses 1**
>
> >What is the end‑to‑end costing (encoder inference + LLM queries): tokens per predicate, latency, and monetary cost per dataset.
>
> Thank you for raising the question regarding the cost.
>
>  The differentiable training is performed by $\gamma$ILP, which allows us to obtain the rules and the constants associated with the variables. We then call LLMs just once to translate the semantics of the predicate placeholders from these image constants into natural language.
>
> From this perspective, $\gamma$ILP is the core component responsible for interpretability, while the LLMs serve only as translators. In our experiments, we used the free user interfaces provided by Google Gemini and GPT.
>
>  Therefore, we consider the number of tokens and the latency (which is very small) to be irrelevant to our experimental objectives.
>
>
>
>
>
>
> >What do you think of learnable truth‑scoring for placeholder atoms instead of a strict logical AND over constant presence?
>
>
> We clarify that $\gamma$ILP learns the truth-score for the atom with a placeholder. For instance, learning from the two-pair task, $\gamma$ILP induces rules with a two-arity predicate placeholder. When learning from the one-red and one-triangle task, $\gamma$ILP induces rules with a one-arity predicate placeholder. We are not learn the logical relations only include the AND over constants, but the AND over atoms with a predicate placeholder.
>
>
> **Weaknesses**
>
> >I believe paper's central claim of a fully differentiable' pipeline relies on a non-differentiable. The predicate invention step, as detailed in Algorithm 2, relies on a non-differentiable, hard query to a frozen LLM. This is the step where the actual conceptual' label (e.g., `triangle') is introduced, and it lies completely outside the end-to-end optimization loop. This architectural choice has implications -- the model does not learn concepts like red' or triangle' through backpropagation. Instead, it learns to form a cluster of constants that happen to be red triangles, and then it asks an external oracle (the LLM) for the appropriate human-readable label. This should be stated explicitly and transparently if my interpretation is correct as it reframes the contribution.
>
> We agree that the LLM-based naming step is non-differentiable; however, this step is not part of predicate invention in the learning sense, nor does it enter the computational graph of $\gamma$ILP. Its sole purpose is to translate to human-readable names for the latent predicates, allowing users to qualitatively assess the learned first-order rules.
>
> We have revised the manuscript for clarity, indicating where the fully rule-learning process begins and ends, specifically at line 53 in Section 1 and line 215 in Section 4.
>
>
> Integrating the LLM into the end-to-end differentiable pipeline would indeed be challenging, as it would require aligning the image-based latent space with a language-based one—a substantially different research problem that falls outside the scope of this work, even if it would be definitely something interesting to work on.

---

> ### Author Response · Authors · 2025-11-21
> **Response to Weaknesses 2 and 3**
>
> >The reported runtimes (Table 4, Appendix) focus on $\gamma$ILP's own training steps but do not discuss the immense computational resources required by ViT, BERT, and especially LLM queries. Training and inferring with these foundation models comes with high cost in time, compute, and financial resources.
>
> We note that $\gamma$ILP does not backpropagate through ViT and BERT, meaning these models do not need to be trained. We use them only for one-time inference to obtain representations of constants and predicates (if predefined). These representations can be stored and reused in subsequent experiments, so the training time for these encoders is not included.
>
> Moreover, using these encoders, such as ViT, for inference is computationally light and feasible in a typical lab setting. Training the perceptual backbone from scratch is also a viable option, as our pipeline already supports backpropagation through these components. Exploring a fully from-scratch variant represents an interesting direction for future work.
>
>
> >There is no holistic accounting of these requirements, nor is the LLM inference cost (token count, latency, etc.) analyzed.
>
> Regarding the use of LLMs, because the prompts are short and the learned constants are represented as variables from $\gamma$ILP, we employ a freely available LLM interface to automatically translate the semantics of predicate placeholders into natural language.
>
>
>
> We note that $\gamma$ILP backpropagates through ViT and BERT, meaning we fine-tune only their last layers, not train them from scratch. This is standard practice (``upstream fine-tuning'') in multimodal and neuro-symbolic pipelines.
>
> Abandoning the upstream setting and training the perceptual backbone from scratch is also a viable option, since our pipeline already backpropagates through these components. However, this would forgo the advantages of leveraging established pretrained models, which is standard practice in multimodal learning. Exploring a fully-from-scratch variant is an interesting direction for future work.
>
>
> For the LLMs usages, because the prompt is short and learned constants are represented by variables from $\gamma$ILP, we use a free LLMs user interface to translate the semantics of the predicate placeholder to natural language automatically.
>
>
>
> >Some experiments appear to encode label cardinality via the number of clusters and rely on pre‑trained encoders with class‑discriminative structure, blurring the “no label leakage” claim.
>
> This question is very similar to the first question from Reviewer 1; please also refer to our response there.
>
> Additionally, we would like to emphasize that our model is built upon several foundational models, such as ViT, BERT, and LLMs. However, it does not require exact symbolic labels, as reasoning is performed over the learned representations.
> The representations of textual relations and image constants can also be generated using variational autoencoders.
>
> The experimental results using a pre-trained variational autoencoder to compute representations of relations and image constants on the relational image dataset are presented below. Specifically, the average recall of rules with a precision of 1 over ten experiments is reported. Here, $\gamma$ILP uses both the autoencoder and ViT on relational images.
>
> | Task | $\gamma$ ILP with VAE | $\gamma$ILP with ViT |
> | --- | --- | --- |
> | Predecessor | 1.00 | 1.00 |
> | Odd | 1.00 | 1.00 |
> | Even | 1.00 | 1.00 |
> | Lessthan | 1.00 | 1.00 |
>
> The recall of the learned rules is 1, indicating we can learn complete rules.
>
> The accuracies obtained using a pre-trained variational autoencoder to compute the representations of image constants on the Kandinsky datasets are presented below:
>
> | Task | $\gamma$ ILP with VAE | $\gamma$ILP with ViT |
> | --- | --- | --- |
> | Two-pair | 0.64 | 0.75 |
> | One-red | 0.77 | 1.00 |
> | One-triangel | 0.77 | 1.00 |
>
> Besides, we use the same pipeline that can also generate interpretable rules, as we have already presented in Section 5.2.
>
> We updated these statements in line 185 of Section 3.3 and line 247 in Section 4.1.

---

> ### Author Response · Authors · 2025-11-21
> **Response to Weakness 4 and 5**
>
> >Failures on classical tasks (Fizz/Buzz) signal a combinatorial bottleneck in bottom‑up propositionalization with many variables/longer chains.
>
>
> We agree that introducing structural constraints can further improve scalability. In fact, we are currently investigating some options, such as imposing a constraint that enforces a tree-shaped structure on rule bodies. This idea is inspired by existential query answering, where acyclicity in the query ensures controlled complexity (see ``Algorithms for acyclic database schemes'' of Yannakakis). We believe such constraints could significantly reduce the search space in our setting as well.
> That said, this extension involves nontrivial design choices and additional theoretical discussion, and we view it as an important direction for future work (mentioned in Section 6) rather than something that can be included in the present submission due to space limitations.
>
>
> >Several results are reported as “best of N runs,” with substantial variance on harder tasks. It will be helpful to see mean±std over fixed seeds and confidence intervals.
>
> To interpret the data using rules, we employ the learned rules corresponding to the most accurate results, focusing exclusively on the Kandinsky dataset.
> For tasks involving Kandinsky patterns, we report the mean ± standard deviation of accuracy over 10 runs, as shown in Figure 12 in the Appendix.
> For the relational image task, the model consistently learns perfect rules, with both recall and precision having a mean of 1 and a standard deviation of 0 across 10 runs.
> Similarly, the traditional ILP model achieves a mean of 1 and a standard deviation of 0 across 10 different seeds.

---

> > ### Comment · Reviewer_dz7p · 2025-11-25
> >
> > I thank the authors for their detailed responses and the additional experiments.

---

> ### Author Response · Authors · 2025-11-26
>
> We appreciate the reviewer’s acknowledgement and believe that we have addressed your concerns thoroughly. We kindly ask you to consider updating your review score, or if needed, we would be happy to discuss further.
>
> Best regards,

---

### Official Review · Reviewer_rXSc · 2025-10-31

**Soundness:** 2
**Presentation:** 2
**Contribution:** 2
**Rating:** 4
**Confidence:** 3

**Summary:**

The paper proposes $\gamma$ILP, a framework designed to learn first-order logic rules from visual data without requiring explicit image labels. The authors claim this approach tackles the symbol grounding problem without "label leakage", enables "predicate invention" for undefined relations, and the proposed pipeline is end-to-end differentiable. The framework is evaluated on classical symbolic ILP datasets, relational image datasets using MNIST, and pure image datasets (Kandinsky patterns).

**Strengths:**

1. The framework uses LLMs to automatically assign semantic meaning to newly discovered relations, a step toward automated knowledge discovery.
2. The authors claim that $\gamma$ILP successfully integrates modern deep learning with a symbolic rule-learning network in a fully differentiable pipeline, enabling joint training.
3. The authors test their method across a diverse set of tasks, from classical symbolic datasets to complex visual reasoning with Kandinsky patterns, demonstrating its versatility.

**Weaknesses:**

1. The core claim of automated predicate invention depends entirely on pre-trained LLMs acting as an oracle. This reliance is unrigorous. Moreover, it essentially reduces to prompt engineering, which is highly sensitive to phrasing, model choice, and randomness.
2. The key generalizations from instance-specific outputs to abstract predicates appear to be manually derived by the authors, wich contradicts the claim of full automation. The LLM contributes only loosely interpreted semantics, introducing subjective human input. This undermines the claimed autonomy of the system.
3. Using an LLM for semantic interpretation introduces non-transparent.  Positioning such a unverifiable framework within explainable AI is conceptually inconsistent.
4. The experiments fail to convincingly validate novelty or effectiveness. Specifically, the baselines are poorly chosen, which compare relational reasoning tasks to non-relational models. The ablations are missing, leaving component contributions unverified.
5. The “differentiable substitution” mechanism is poorly explained and insufficiently contrasted with prior work.

**Questions:**

Does the claim of a “fully differentiable pipeline” hold, given that predicate invention depends on a non-differentiable LLM query?

---

> ### Author Response · Authors · 2025-11-21
> **Response to Question and Weakness 1**
>
> Thank you for your comments. We have updated the manuscript accordingly. Please find our detailed responses below.
>
> **Questions**
>
> >Does the claim of a ``fully differentiable pipeline'' hold, given that predicate invention depends on a non-differentiable LLM query?
>
>
> Yes, the claim of a ‘fully differentiable pipeline’ holds. We use a fully differentiable pipeline to transform raw data into symbolic rule structures, as mentioned in the Abstract. The LLM is not part of the computational graph; it is applied only after training to translate the invented latent predicates, represented by the selected image constants, into human-readable names.
>
> Predicate invention—that is, discovering clusters, grounding variables, and introducing new predicates—is performed entirely by the differentiable model $\gamma$ILP. All parameters involved in this process, both during learning and inference, receive gradients.
>
> The rule structures learned by $\gamma$ILP include the invented predicate placeholders, the ordering of clusters, and the images associated with each cluster. We can interpret the semantics of the predicates by examining the images within each cluster.
>
> The LLM query is therefore not in the loop of learning, or inference, and removing it would not change the learned model in any way.
> To avoid misunderstanding, we can clarify in the final version that the LLM component is optional and purely cosmetic, with no effect on differentiability or end-to-end learning. Please see the updates at Line 54 in Section 1.
>
> **Weaknesses**
> >The core claim of automated predicate invention depends entirely on pre-trained LLMs acting as an oracle. This reliance is unrigorous.
>
> We would like to clarify that the core automated predicate invention does not
> depend on the pre-trained LLMs, but on our differentiable model, $\gamma$ILP.
>
> As we explain in the question, $\gamma$ILP learns how to map images to clusters (constrained variables) under unary or binary predicates; in other words, it learns how to ground the program by discovering the latent concepts. This constitutes the actual predicate invention.
> The LLM is used only to translate human-readable
> names from these invented predicates for qualitative inspection.
>
> >Moreover, it essentially reduces to prompt engineering, which is highly sensitive to phrasing, model choice, and randomness.
>
> This naming phase occurs after the learning process and does not affect the learned rules, the discovered concepts, or the final predictions. Using the rules and constants within clusters learned by $\gamma$ILP, we employ a simple and deterministic prompt to translate the unary or binary concepts identified by $\gamma$ILP, without any additional prompt engineering.
>
> We also compare the sensitivity of choosing different LLMs to translate the semantics.
> We chose GPT-5, GPT-3o, and Gemini 2.5 Pro to compare the translation ability from the output of $\gamma$ILP. For the two-pair task, we use the prompt:
> 'What is the relation between images under cluster R and the images under cluster V?' and test with the images under clusters $R$ and $V$ Figure 2 (d).
> For the one-red task, we use the prompt: 'What is the common property of these objects in the image?' and test with the image under the cluster $U$ in Figure 3 (a). For the one-triangle task, we use the prompt:' What is the common property of these objects in the image?' and test with the image in Figure 3 (b). The results are:
>
> | Tasks | GPT-5 | GPT-3o | Gemini 2.5 Pro |
> | --- | --- | --- | --- |
> | Two-pair | Cluster R and Cluster V represent the same shape category, but differ in color. | Everything about the paired images is identical—shape, size, cropping, background—except for the hue of the triangle. | The relationship is one of contrast in the color dimension, while maintaining consistency in the shape and overall format. |
> | One-red | They are all solid red geometric shapes. | Every item in that set is filled with the same solid red colour; the hue is the property they all share. | color (red) is the property they all possess in an identical state, making it the primary commonality. |
> | One-triangle  | All the objects in the image share one clear common property: They are all triangles. | Every one of the thumbnails shows the same basic geometric shape: an upright, isosceles (appearance-wise almost equilateral) triangle. Color, slight aliasing and exact position vary, but the common property is simply that they are all triangles pointing upward inside the square frame. | Based on the provided image, the most immediate and common property of all the distinct triangular objects shown is that they are all isosceles triangles. |
>
> The experimental results indicate that the semantics can be easily inferred by all considered LLMs. Additionally, GPT-5, GPT-3o, and Gemini 2.5 Pro can translate the same semantics under each rule learned by $\gamma$ILP.
> Please also see the same results presented in Table 5, Appendix D of the updated manuscript.

---

> ### Author Response · Authors · 2025-11-21
> **Response to the Remaining Weaknesses**
>
> >The key generalizations from instance-specific outputs to abstract predicates appear to be manually derived by the authors, which contradicts the claim of full automation. The LLM contributes only loosely interpreted semantics, introducing subjective human input. This undermines the claimed autonomy of the system.
>
> The rule-learning process generated by $\gamma$ILP is highly precise in capturing predicate semantics. However, these semantics are encoded within the cluster order and the constants represented by the clusters, rather than explicitly labeled. Humans can interpret them in natural language. To automate this interpretation, we use an LLM as a translator to convert the semantics hidden in the variables into natural language.
>
> We randomly select 20 constants from the knowledge base and further examine the constants within each cluster, which are fine-tuned by the rule-learning module during training. From these constants, we can extract the abstract predicate semantics. Importantly, the abstract predicates are not manually chosen by humans.
>
>
> >Using an LLM for semantic interpretation introduces non-transparent. Positioning such a unverifiable framework within explainable AI is conceptually inconsistent.
>
> The LLM is not part of the reasoning or learning pipeline; it is used solely to assign optional human-readable names to the latent predicates already discovered by $\gamma$ILP. Specifically, interpretability is achieved through the rules and the constants represented by the variables within those rules. By analyzing the rules and the constants bound to the clusters, we can derive the interpretability of the learned model.
>
>
> >The experiments fail to convincingly validate novelty or effectiveness. Specifically, the baselines are poorly chosen, which compare relational reasoning tasks to non-relational models.
>
>
> We believe our experiments are thorough and the comparisons with baselines are conducted fairly. Specifically, for the conventional learning setting using ILP datasets, we compare $\gamma$ILP with state-of-the-art models such as DFORL, $\partial$ILP, and the leading LLMs with the strongest reasoning capabilities.
>
> When learning from relational datasets with image constants, existing models such as $\alpha$ILP and $\partial$ILP require symbolic labels for the constants as input. In contrast, $\gamma$ILP learns rules without these labels. Therefore, we compare its accuracy with state-of-the-art LLMs that also do not require label input.
>
> For learning from pure image datasets, we evaluate the accuracy of $\gamma$ILP and compare it with leading visual models, including CNN, ViT, YOLO, and language models. The results indicate that $\gamma$ILP achieves the best performance.
>
> Across all these experiments, $\gamma$ILP not only delivers excellent accuracy but also provides clear interpretability through the rules it generates.
>
> We also note the reviewer’s comment regarding comparisons with non-relational models, such as RIPPER or C4.5, in the Appendix. As stated in the manuscript, even though RIPPER and C4.5 can be combined with encoders and clustering methods, the learned rules lack the level of interpretability offered by the first-order rule learning module integrated with an encoder and clustering. Additional details can be found at line 1096 in Appendix E.
>
>
> >The ablations are missing, leaving component contributions unverified.
>
> We have conducted ablation studies in Appendix E, examining the model’s sensitivity to hyperparameters such as the clustering learning rate, the number of clusters, the learning rate of the rule learning module, and the values of $\alpha$ and $\lambda$. Regarding baselines and ablations, we would be happy to include additional ablation studies based on any further explicit comments from the reviewer.
>
>
>
>
> >The ``differentiable substitution'' mechanism is poorly explained and insufficiently contrasted with prior work.
>
> The ‘differentiable substitution’ mechanism is indeed novel in this work; to the best of our knowledge, no prior differentiable ILP or neuro-symbolic system performs variable–value substitution through a learnable soft grounding mechanism in this manner.
>
> We explicitly describe differentiable substitution in Section 4.2. Specifically, Section 3.1 introduces the definition of substitution, which are prepared for substitution depending on whether the relations are defined or undefined, as discussed in Section 4.2. Algorithm 1 details the substitution method when the relations are defined, including its inputs and outputs. Finally, we describe how substitution is performed when the relations are not predefined, which is necessary for learning from Kandinsky patterns.
>
> In the revised manuscript, we have added further explanation regarding substitutions under undefined relations at line 310 in Section 4.2.

---

> > ### Comment · Reviewer_rXSc · 2025-11-22
> >
> > I appreciate the authors’ detailed response. However, the rebuttal version of the manuscript does not appear to highlight the revisions. Please add difference highlights to help distinguish the changes.

---

> > > ### Author Response · Authors · 2025-11-22
> > >
> > > Thank you very much for your prompt response. We have uploaded a rebuttal version with highlights in the supplementary materials to indicate the revised content.
> > >
> > > In this revision, we have mainly updated the manuscript based on the reviewers’ valuable comments, and we have also adjusted the notations to ensure greater consistency.
> > >
> > > Please feel free to let us know if you have any concerns. We look forward to your further feedback.

---

> ### Author Response · Authors · 2025-11-27
>
> Dear Reviewer,
>
> We hope that we have addressed your concerns and questions thoroughly.
> We would be grateful if you could consider updating your evaluation, or let us know if further clarification or discussion is needed.
>
> Best regards,
>
> The Authors

---

### Official Review · Reviewer_NaeU · 2025-10-31

**Soundness:** 3
**Presentation:** 3
**Contribution:** 3
**Rating:** 6
**Confidence:** 3

**Summary:**

The paper proposes γILP, a fully differentiable inductive logic programming (ILP) framework that provides a fully differentiable pipeline from entity substitution to rule structure induction. This method consists of a deep clustering module serving as a generalization function, a latent knowledge base generator, a rule learning neural network with a novel differentiable substitution method, and LLMs acting as concept learners. Also,It is evaluated on classical symbolic ILP tasks, relational image datasets (e.g., MNIST-based relational reasoning), and pure image tasks (Kandinsky patterns), showing competitive performance and interpretative rule induction.

**Strengths:**

1.	No label leakage: the first model to learn rules from relational image datasets without label leakage which is a key limitation in prior ILP work.
2.	Broad empirical validation: across diverse tasks (symbolic, relational images, Kandinsky) shows robustness.
3.	Interpretative predicate invention: Predicate invention via LLMs is a practical and interpretative solution to a classic ILP challenge.

**Weaknesses:**

1.	LLM dependency: Predicate invention is outsourced to black-box LLMs; no analysis of prompt sensitivity or LLM choice.
2.	Task limits: The method struggles with tasks (e.g., Fizz/Buzz), “In γILP, the learned rule can only describe the entity zero, but fails to capture numbers divisible by three and five in the Fizz and Buzz datasets”, “the search space for γILP and DFORL becomes enormous without constraints such as the logical template used in ∂ILP”, suggesting limited scalability in complex relational reasoning without strong inductive biases or templates.

**Questions:**

1.	The paper emphasizes that γILP is “the first model to learn rules from relational image datasets without label leakage”. However, it relies on pre-trained ViT and BERT encoders that were trained on large labeled datasets. Given that these encoders may already embed semantic distinctions between colors and shapes—even without task-specific labels—could this constitute a form of implicit label leakage through the perceptual backbone?
2.  Recently, there is an increasing number of  studies addressing relational imaging learning such as, raven progressive matrices. I am not sure how this model can learned abstract rules for solving raven reasoning tasks.
3.	The paper mentions in Section 5.2 (p. 8, lines 456–458) that the learned rule for the two-pair Kandinsky task achieves recall = 1 but precision < 1 because it fails to account for another pair of entities with the same color and shape. Since γILP fixes the number of logic variables to the number of clusters (Section 4.3, p. 6), might this design choice restrict the model’s performance for tasks that require reasoning over multiple relational pairs simultaneously


References:
1. https://arxiv.org/abs/1903.02741

---

> ### Author Response · Authors · 2025-11-21
> **Response to Question 1**
>
> We thank you for your support and the valuable comments on our work. Please find our detailed responses to the questions and identified weaknesses below.
>
> **Questions**
>
> > The paper emphasizes that $\gamma$ILP is ``the first model to learn rules from relational image datasets without label leakage''. However, it relies on pre-trained ViT and BERT encoders that were trained on large labeled datasets. Given that these encoders may already embed semantic distinctions between colors and shapes—even without task-specific labels—could this constitute a form of implicit label leakage through the perceptual backbone?
>
> Thank you for the question, as it is a crucial aspect of our work.
> Based on the definition of label leakage as defined in Section 3.2. The symbol grounding problem targets establishing a mapping from a raw image or text input $x$ via its embedding $\mathbf{e} \in \mathbb{E}$ into some latent state $\mathbf{z} \in \mathbb{Z}$ that is fed into a
> differentiable symbolic reasoning procedure for producing the final output $y$. The label leakage indicates that we will not provide the label information for reasoning regarding the input $x$.
>
> BERT and ViT are indeed pre-trained models. In our learning setup, we use only the embeddings of the constants generated by these foundation models, such as ViT and BERT, without providing any symbolic labels for the constants. Consequently, we avoid any form of label leakage, according to the standard definition based on a model’s inputs and outputs. As foundation models, BERT and ViT are designed to support downstream tasks, and we believe that leveraging their pre-trained representations can meaningfully benefit neuro-symbolic learning by improving generalization.
>
> Technically, when predicates are predefined and constants are textual, any encoder can be used to embed the predicate symbols. $\gamma$ILP uses these predicate embeddings only during the lookup stage for representation and value checking. No clustering is performed on the representations for textual data. Hence, using any of the encoders for textual predicates or constants won't change the performance of $\gamma$ILP.
>
> For constants represented as images, we require similar images to have similar representations to train the clustering module effectively. Therefore, in addition to using ViT, we also train a variational autoencoder (VAE) on these constant images and use this pretrained VAE to generate embeddings of the image constants when learning from relational images and Kandinsky patterns.
>
> The new experimental results are presented below.
> We ran the experiments ten times and computed the average recall of the learned rules, with the precision fixed at 1. The results show that $\gamma$ILP, using both the autoencoder and ViT, achieves a mean recall of 1 on the relational image task.
>
> | Task | $\gamma$ ILP with VAE | $\gamma$ILP with ViT |
> | --- | --- | --- |
> | Predecessor | 1.00 | 1.00 |
> | Odd | 1.00 | 1.00 |
> | Even | 1.00 | 1.00 |
> | Lessthan | 1.00 | 1.00 |
>
> The recall of the learned rules is 1, indicating that the model successfully learns a complete set of rules to describe the target atom.
>
> The best accuracies with ten runs with the autoencoder on the Kandinsky datasets are presented in the following table:
> | Task | $\gamma$ ILP with VAE | $\gamma$ILP with ViT |
> | --- | --- | --- |
> | Two-pair | 0.64 | 0.75 |
> | One-red | 0.77 | 1.00 |
> | One-triangel | 0.77 | 1.00 |
>
> Besides, when we use $\gamma$ILP with an autoencoder, we can also generate interpretable rules under the best accuracy, as we have presented in Section 5.2.
>
> We also acknowledge that $\gamma$ILP combined with a ViT encoder achieves the highest mean accuracy on the Kandinsky patterns. However, regardless of the chosen encoder, $\gamma$ILP does not require symbolic labels as inputs. This flexibility highlights the strong modularity and representational adaptability of our approach.
>
> We have integrated these statements and experimental results
> in Section 4.1, Table 1, and Table 2 in the revised version of
> the manuscript.

---

> ### Author Response · Authors · 2025-11-21
> **Response to Remaining Questions and Weakness 1**
>
> >Recently, there is an increasing number of studies addressing relational imaging learning such as, raven progressive matrices. I am not sure how this model can learned abstract rules for solving raven reasoning tasks.
>
>
> Thank you for pointing out the Raven reasoning tasks.
>
>  When considering the use of $\gamma$ILP to solve Raven-style reasoning problems, we note that these datasets contain temporal or spatial relations between different constants. Our current model abstracts only the representations of the constants through encoders, without incorporating temporal or spatial information. Nevertheless, we believe that incorporating spatial relations between constants is a promising direction for future work. We have added this discussion and cited the referenced work in the Conclusion section of the revised manuscript.
>
> [1] Chi Zhang, Feng Gao, Baoxiong Jia, Yixin Zhu, Song-Chun Zhu: RAVEN: A Dataset for Relational and Analogical Visual Reasoning. CVPR 2019: 5317-532
>
> > The paper mentions in Section 5.2 (p. 8, lines 456–458) that the learned rule for the two-pair Kandinsky task achieves recall $= 1$ but precision $< 1$ because it fails to account for another pair of constants with the same color and shape. Since $\gamma$ILP fixes the number of logic variables to the number of clusters (Section 4.3, p. 6), might this design choice restrict the model’s performance for tasks that require reasoning over multiple relational pairs simultaneously
>
> Thank you for the suggestion. We have also considered this issue.
> At present, we interpret the predicate semantics based on the arity of the predicate placeholder, the ordering of the cluster indices, and the constants contained within each cluster. Each cluster corresponds to a constrained variable, and we generalize the rules using these constrained variables.
>
> If we remove the requirement that the number of constrained variables matches the number of clusters in $\gamma$ILP, the interpretability would be lost.
> However, during training, we can set the number of constrained variables to 10, which is sufficiently large to allow learning diverse rules while preserving interpretability.
>
> During the learning process, $\gamma$ILP may produce multiple rules. In particular, we obtain a rule shown in Equation (11) in Appendix C. This rule has a precision of 1 but a recall of less than 1; it correctly captures the two-pair semantics but only for a specific pattern.
> Based on this rule and the constants within the corresponding clusters, we can interpret it as follows: two squares share the same shape but differ in color, while the remaining patterns share both the same color and the same shape.
>
> As future work, we will consider your suggestion to update the neural network architecture and further improve both the precision and recall of the rules in the Kandinsky two-pair task.
>
>
>
> **Weaknesses**
>
> > LLM dependency: Predicate invention is outsourced to black-box LLMs; no analysis of prompt sensitivity or LLM choice.
>
> Thank you for the helpful comments.
>
>  We would like to clarify the role of LLMs in our work.
> First, we do not consider the use of LLMs to be a core contribution, as they play no role in the learning or inference pipeline; we will emphasize this more clearly in the final version of the manuscript.
> The results produced by the differentiable pipeline $\gamma$ILP specify the predicate placeholder and the images represented by the constrained variables (clusters) in the learned logic rules. For example, in $Positive \leftarrow p(Z)$, $\gamma$ILP learns the specific images associated with the constrained variable $Z$.
> Thus, predicate invention is already accomplished at this stage by interpreting the semantics of the selected images through $\gamma$ILP.
> To enhance human readability, we use an LLM solely to translate a meaningful name to $p$ based on the images represented by $Z$. This step is optional and not required for learning the rules or performing inference.
> We have updated these statements at Line 54 in Section 1 of the manuscript.
>
>
> Because the clustering method is fine-tuned by the rule learning module, the constants within the same cluster may not always share the same shape and color; for example, see the constants in the cluster shown in Figure 3(b).
> Since all input information is explicit and the LLM serves only as a verification tool to translate the semantics of predicate placeholders in natural language, we do not consider prompt sensitivity, which may break the output of $\gamma$ILP. Instead, we use a simple, specific, and well-designed prompt to translate the image-based semantics into natural language.

---

> ### Author Response · Authors · 2025-11-21
> **Response to Weakness 2**
>
> > Task limits: The method struggles with tasks (e.g., Fizz/Buzz), `In $\gamma$ILP, the learned rule can only describe the constant zero, but fails to capture numbers divisible by three and five in the Fizz and Buzz datasets'', ``the search space for $\gamma$ILP and DFORL becomes enormous without constraints such as the logical template used in $\delta$ILP'', suggesting limited scalability in complex relational reasoning without strong inductive biases or templates.
>
> We agree that introducing structural constraints can further improve scalability. In fact, we are currently investigating some options, such as imposing a constraint that enforces a tree-shaped structure on rule bodies. This idea is inspired by existential query answering, where acyclicity in the query ensures controlled complexity (see ``Algorithms for acyclic database schemes'' of Yannakakis). We believe such constraints could significantly reduce the search space in our setting as well.
> That said, this extension involves nontrivial design choices and additional theoretical discussion, and we view it as an important direction for future work (mentioned in Section 6) rather than something that can be included in the present submission due to space limitations.

---

> > ### Comment · Reviewer_NaeU · 2025-11-27
> >
> > I thank the authors' responses. It sounds OK to me. I decide to keep my score.

---

> > > ### Author Response · Authors · 2025-11-27
> > >
> > > We appreciate your support.
> > >
> > > Best regards,

---

### Author Response · Authors · 2025-11-30
**Message to AC**

Dear AC,

Firstly, we would like to thank you for your effort in reviewing our submission to ICLR.

We would like to highlight that we have thoroughly addressed each question raised by all the reviewers. Specifically, the first and third reviewers (NaeU and dz7p) have acknowledged our responses to their questions and the revised manuscript, which includes clearer presentations based on their suggestions and identified weaknesses.


In addition to their agreement with our contributions, we would like to highlight that some comments from the reviewers are similar.
For instance:
- Weakness 1 proposed by Reviewer 1 (NaeU),
- Question and Weaknesses 1, 2, and 3 proposed by Reviewer 2 (rXSc),
- Questions 1 and 3, and Weakness 1 proposed by Reviewer 3 (dz7p), and
- Question proposed by Reviewer 4 (AuPA),
all share similar concerns.

We have addressed this by clarifying that predicate invention is carried out by $\gamma$ILP, not by large language models (LLMs). Additionally, the predicate invention process is fully automated. The LLMs are only used to translate the invented semantics of the predicates by $\gamma$ILP into a more readable natural language format. LLMs here are used in a very light way by their user interface.

Additionally, Weakness 3 proposed by Reviewer 3 (dz7p) and Question 1 proposed by Reviewer 1 (NaeU) are similar.
We would like to clarify that we do not finetune any encoders during the learning process. Furthermore, we have tested $\gamma$ILP's performance using a variational autoencoder, which further verifies that our learning process does not rely on explicit labels as input.

Furthermore, we perform multiple validation experiments to test the sensitivity of $\gamma$ILP under different hyperparameters, such as the number of clusters and the value of the learning rate.
In addition, we compare $\gamma$ILP with the state-of-the-art reasoning model under the same settings, without using label information, across multiple running times. We present the mean accuracy and standard deviations to show the performance consistency.

In conclusion, $\gamma$ILP learns easily understandable rules to describe raw images without relying on human-defined logic templates or image labels (thus avoiding label leakage), and it learns the rules solely on GPUs in a differentiable manner.
During the learning process, predicates can be invented by the learned rules, the order of variables, and the image constants represented by these variables in the rules.
The experimental results indicate that our model achieves state-of-the-art performance on the Kandinsky tasks.

We hope our contributions, particularly the first work featuring predicate invention, symbol grounding without label leakage, and fully GPU-based learning of symbolic rules, will make a valuable contribution to the ICLR 2026 conference.


Best regards,

Authors

---

### Meta-Review · Area_Chair_CpcD · 2026-01-04

**Summary:**

This paper proposes a differentiable neuro-symbolic framework for learning first-order logic rules from images without explicit labels. Reviewer enthusiasm was limited. Scores clustered around borderline (two 6s and two 4s), and importantly, no reviewer clearly advocated for acceptance. The main concerns were the strength/clarity of the core claims, scalability, and whether the empirical evidence is sufficient beyond controlled benchmarks.

**Reviewer Concerns:**

Reviewer concerns:

** partially addressed **
- clarification that LLMs are used only for post-hoc naming and are not part of the differentiable learning pipeline
- additional explanations and experiments related to pretrained encoders and the “no label leakage” framing
- some additional reporting of mean and standard deviation on Kandinsky-style tasks

** still outstanding **
- scalability limitations remain, including failures on tasks like Fizz/Buzz and reliance on future constraints to control combinatorial search
- “no label leakage” claim remains debatable, given reliance on pretrained encoders with strong semantic priors and choices like the number of clusters
- concerns about experimental rigor and positioning, including baseline selection, limited ablations/component verification, and incomplete contrast to prior differentiable neuro-symbolic work
- end-to-end compute/cost accounting is not convincingly quantified

** AC concerns **
- the paper is close to the borderline in its current form and does not make a compelling case for acceptance in a competitive venue without a strong advocate in the reviews
- results are promising on controlled settings but do not convincingly establish robustness or scalability

**Reviewer Scores:**

- reviewer NaeU: 6, unchanged
- reviewer rXSc: 4, likely unchanged
- reviewer dz7p: 4, possibly slightly higher but still below threshold
- reviewer AuPA: 6, likely unchanged

---

### Decision · Program_Chairs · 2026-01-26

Reject